# Transitioning between preparatory and precisely sequenced neuronal activity in production of a skilled behavior

Vamsi K Daliparthi[1], Ryosuke O Tachibana[2,3], Brenton G Cooper[4], Richard HR Hahnloser[3,5], Satoshi Kojima[6], Samuel J Sober[7], Todd F Roberts[1]*

[1]Department of Neuroscience, UT Southwestern Medical Center, Dallas, United States; [2]Department of Life Sciences, The University of Tokyo, Tokyo, Japan; [3]Institute of Neuroinformatics, University of Zurich/ETH Zurich, Zurich, Switzerland; [4]Department of Psychology, Texas Christian University, Fort Worth, United States; [5]Neuroscience Center Zurich (ZNZ), Zurich, Switzerland; [6]Department of Structure and Function of Neural Network, Korea Brain Research Institute, Daegu, Republic of Korea; [7]Department of Biology, Emory University, Atlanta, United States

**Abstract** Precise neural sequences are associated with the production of well-learned skilled behaviors. Yet, how neural sequences arise in the brain remains unclear. In songbirds, premotor projection neurons in the cortical song nucleus HVC are necessary for producing learned song and exhibit precise sequential activity during singing. Using cell-type specific calcium imaging we identify populations of HVC premotor neurons associated with the beginning and ending of singing-related neural sequences. We characterize neurons that bookend singing-related sequences and neuronal populations that transition from sparse preparatory activity prior to song to precise neural sequences during singing. Recordings from downstream premotor neurons or the respiratory system suggest that pre-song activity may be involved in motor preparation to sing. These findings reveal population mechanisms associated with moving from non-vocal to vocal behavioral states and suggest that precise neural sequences begin and end as part of orchestrated activity across functionally diverse populations of cortical premotor neurons.
DOI: https://doi.org/10.7554/eLife.43732.001

*For correspondence:
todd.roberts@utsouthwestern.edu

## Introduction

The sequential activation of neurons is implicated in a wide variety of behaviors, ranging from episodic memory encoding and sensory processing to the voluntary production of skilled motor behaviors (*Fee et al., 2004*; *Fiete et al., 2010*; *Hahnloser et al., 2002*; *Li et al., 2015*; *Lynch et al., 2016*; *Markowitz et al., 2015*; *Okubo et al., 2015*; *Peters et al., 2014*; *Rajan et al., 2016*; *Svoboda and Li, 2018*). Neural sequences develop through experience and have been described in several brain areas, including the motor cortex, hippocampus, cerebellum, and the basal ganglia (*Barnes et al., 2005*; *Dragoi and Buzsáki, 2006*; *Foster and Wilson, 2006*; *Harvey et al., 2012*; *Jin et al., 2009*; *Li et al., 2015*; *Luczak et al., 2007*; *Mauk and Buonomano, 2004*; *Peters et al., 2014*; *Pfeiffer and Foster, 2013*; *Pfeiffer and Foster, 2015*; *Schwartz and Moran, 1999*). Although computational models provide important insights into circuit architectures capable of sustaining sequenced activity (*Churchland et al., 2010b*; *Fiete et al., 2004*; *Fiete et al., 2010*; *Haga and Fukai, 2018*; *Harvey et al., 2012*; *Kumar et al., 2010*; *Rajan et al., 2016*), our understanding of sequence initiation and termination is still limited.

The precise neural sequences associated with birdsong may provide a useful biological model for examining this issue. Premotor projection neurons in the cortical vocal region HVC (HVC$_{RA}$ neurons, see *Figure 1* legend for anatomical abbreviations) exhibit precise sequential activity during song and current evidence suggests that this activity is acutely necessary for song production (*Hahnloser et al., 2002*; *Kozhevnikov and Fee, 2007*; *Long and Fee, 2008*; *Long et al., 2010*; *Scharff et al., 2000*). HVC$_{RA}$ neurons are thought to only be active during vocal production in waking adult birds, yet ~50% of recorded HVC$_{RA}$ neurons do not exhibit any activity during singing (*Hahnloser et al., 2002*; *Hamaguchi et al., 2016*; *Kozhevnikov and Fee, 2007*; *Long et al., 2010*; *Lynch et al., 2016*), leaving the function of much of the HVC$_{RA}$ circuitry unresolved.

Neuronal activity related to motor planning and preparation has been associated with accurate production of volitional motor movements (*Churchland et al., 2010a*; *Svoboda and Li, 2018*) but is still poorly described in the context of initiating precise neural sequences for motor behaviors, like those exhibited in HVC$_{RA}$ neurons. Although it is not known if HVC$_{RA}$ neurons exhibt activity related to motor planning and preparation, previous studies have identified anticipatory or preparatory activity in other classes of HVC neurons and in other regions of the songbird brain (*Goldberg et al., 2010*; *Goldberg and Fee, 2012*; *Kao et al., 2008*; *Keller and Hahnloser, 2009*; *Rajan, 2018*; *Roberts et al., 2017*). HVC contains interneurons and at least three classes of projection neurons, including neurons projecting to the striatopallidal region Area X (HVC$_X$), neurons projecting to a portion of the auditory cortex termed Avalanche (HVC$_{Av}$), and the aforementioned HVC$_{RA}$ neurons that encode precise premotor sequences necessary for song production (*Akutagawa and Konishi, 2010*; *Mooney and Prather, 2005*; *Roberts et al., 2017*). Multi-unit recordings from HVC, which are typically dominated by the activity of interneurons, show increases in activity tens to hundreds of milliseconds prior to singing (*Crandall et al., 2007*; *Day et al., 2009*; *Rajan, 2018*). Calcium imaging from HVC$_{Av}$ neurons and electrophysiological recordings from HVC$_X$ neurons indicate that they also become active immediately prior to song onset (*Rajan, 2018*; *Roberts et al., 2017*). These data are consistent with recordings from the downstream targets of HVC$_{Av}$ and HVC$_X$ neurons. Portions of the auditory cortex (*Keller and Hahnloser, 2009*) and the basal ganglia pathway involved in song learning show changes in activity immediately prior to singing (*Goldberg et al., 2010*; *Goldberg and Fee, 2012*; *Kao et al., 2008*). Given this background, and that ~50% of HVC$_{RA}$ neurons may not exhibit any activity during singing (*Hamaguchi et al., 2016*; *Long et al., 2010*), we sought to examine if the precise neural sequences associated with song arise as part of larger changes in activity among populations of HVC$_{RA}$ neurons.

To examine the neural circuit activity associated with the initiation and termination of singing, we imaged from populations of HVC$_{RA}$ neurons in freely singing birds. We show that ~ 50% of HVC$_{RA}$ neurons are active during periods associated with preparation to sing and recovery from singing and that their activity presages the volitional production of song by 2–3 s. One population of HVC$_{RA}$ neurons is only active immediately preceding and following song production, but not during either singing or non-vocal behaviors. A second population of neurons exhibits ramping activity before and after singing and can also participate in precise neural sequences during song performance. Recordings from downstream neurons in the motor cortical nucleus RA reveal neural activity prior to song initiation and following song termination. The control of respiratory timing is essential for song (*Schmidt and Goller, 2016*), and our measurements of respiratory activity suggest that one function of pre-singing activity in HVC$_{RA}$ neurons is to coordinate changes in respiration necessary for song initiation. From these findings, we reason that subpopulations of HVC$_{RA}$ neurons are involved in motor planning and motor preparation, encoding the neural antecedents of song that drive recurrent pathways through the brainstem to prepare the motor periphery for song production.

## Results

### Activity sequences in populations of HVC$_{RA}$ neurons

We used miniscope calcium imaging to examine the activity of populations of HVC$_{RA}$ neurons in singing zebra finches (*Chen et al., 2013*; *Ghosh et al., 2011*). A total of 223 HVC$_{RA}$ neurons were imaged during production of 1298 song syllables from six birds (30 song phrases across 18 imaging trials, *Supplementary file 1*). To selectively target HVC$_{RA}$ neurons, we combined retrograde viral

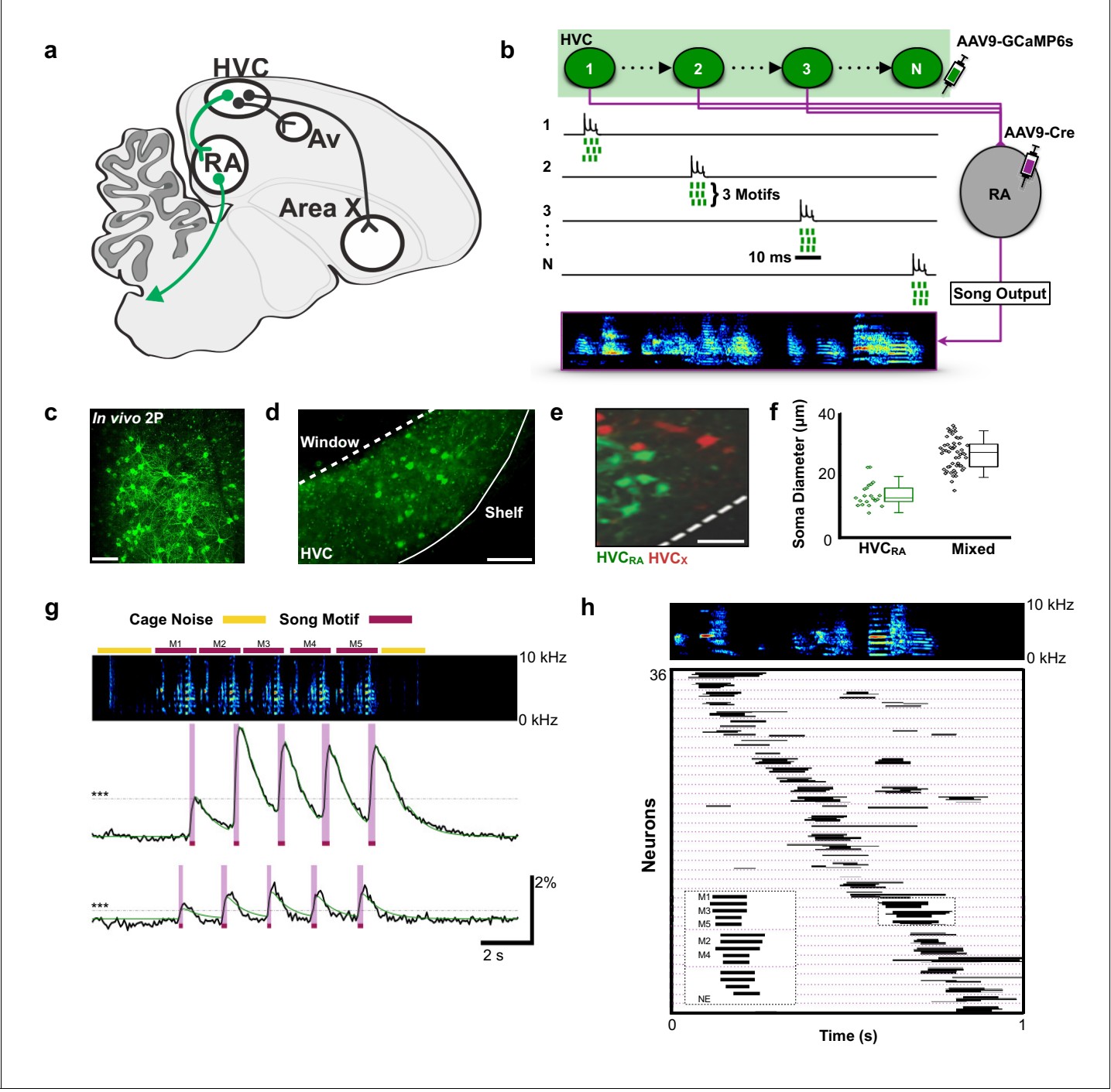

**Figure 1.** Population imaging of song-related HVC_RA sequences. (a) Diagram showing three distinct projection neuron targets of the vocal premotor nucleus HVC. The projection neurons connecting HVC to the downstream motor nucleus RA (HVC_RA neurons) are shown in green. Auditory region Avalanche (Av), robust nucleus of the arcopallium (RA), nucleus HVC of the nidopallium (HVC). (b) Schematic showing HVC_RA neuron somata (green) and their outputs (magenta) to the downstream motor nucleus RA. AAV9-Flex-CAG-GCaMP6s was injected into HVC (green syringe) and AAV9-CAG-Cre was injected into RA (magenta syringe) to selectively label HVC_RA neurons. (c) In vivo two-photon maximum density projection of retrogradely labeled HVC_RA neurons expressing GCaMP6s. Scale bar = 100 µm. (d) A cross-section of HVC showing GCaMP6s-labeled HVC_RA neurons (green). The dashed line indicates where the cranial window was made over HVC. Note the lack of labeling in the region directly ventral of HVC, known as the HVC shelf. Scale bar = 100 µm. (e) A sagittal section of HVC showing HVC_RA neurons (green) and retrogradely labeled HVC_X neurons (red). The dashed line indicates the border between HVC and HVC shelf. Scale bar = 50 µm. (f) Whisker and scatter plots of soma diameters of GCaMP6s expressing cells show that retrogradely labeled HVC_RA neurons (green) have smaller diameters than neurons labeled using only direct viral injections (AAV9-CAG-

*Figure 1 continued*

GCaMP6s) into HVC (mixed population neurons, black). Boxes depict 25th and 75th percentiles, whiskers depict SD. $HVC_{RA}$: N = 21 neurons; mean diameter = 14.0 ± 3.8 μm (SD); Mixed: N = 52 neurons; mean = 26.9 ± 4.9 μm; $t = -10.7$, p=2.0×10$^{-16}$, two-sample $t$ test. (g) Example calcium traces from 2 $HVC_{RA}$ neurons in a bird that sang five consecutive motifs. Shown are the background-subtracted traces (black) and the inferred calcium traces (green). The magenta overlays indicate the rise time (intervals between onset and peak times) of the recorded calcium transients. The horizontal dashed line (gray) denotes 3 SD above baseline activity. The bars above the spectrogram denote cage noise associated with birds hopping or flapping their wings (yellow) or production of song motifs (red). (h) Motif-related activity of 36 $HVC_{RA}$ neurons across five motifs. Each row shows activity of a neuron from one trial. The dashed magenta lines separate different neurons. Empty spaces indicate trials wherein neurons were not active (no event, NE). The inset shows a zoom-in of activity from three separate $HVC_{RA}$ neurons.

DOI: https://doi.org/10.7554/eLife.43732.002

The following source data and figure supplement are available for figure 1:

**Source data 1.** Raw soma diameter measurements for *Figure 1F*.

DOI: https://doi.org/10.7554/eLife.43732.004

**Figure supplement 1.** Diagram showing three distinct projection neuron targets of the vocal premotor nucleus HVC.

DOI: https://doi.org/10.7554/eLife.43732.003

expression of cre recombinase from bilateral injections into RA with viral expression of cre-dependent GCaMP6s from injections into HVC (*Figure 1a–b* and legend, *Figure 1—figure supplement 1*; see Materials and methods) (*Chen et al., 2013*). We confirmed the identity of imaged neurons using conventional retrograde tracing, anatomical measures of neuronal features, and post-hoc histological verification. Although this did not label all RA projecting neurons in HVC, we found that this approach exclusively and uniformly labeled populations of $HVC_{RA}$ neurons (*Figure 1c–f*).

To elicit courtship singing, we presented male birds with a female and imaged $HVC_{RA}$ neurons during song performance (*Video 1*, see Materials and methods for definitions song). On average, birds engaged in singing in 24 s (±49 s) of a female bird being presented. Given the slow decay times of calcium signals relative to singing behavior, we defined neuronal activity by the rise times of calcium events that were >3 standard deviations (SD) above baseline (*Figure 1g*, average rise time: 0.112 ± 0.047 s SD, see Materials and methods). The activity of individual $HVC_{RA}$ neurons was time-locked to a moment in the birds' song, with different neurons active at different moments in the song motif (*Figure 1g–h*; onset jitter = 55.0 ± 60.9 ms, populations imaged at 30 frames per second). We found that the sequential activity of $HVC_{RA}$ neurons roughly coded for all moments in the song motif (*Figure 1h*). These results provide the first glimpse of activity across populations of identified $HVC_{RA}$ neurons during singing and support the idea that sparse and precise neuronal sequences underlie the sequential structure of birdsong (*Amador et al., 2013*; *Hahnloser et al., 2002*; *Long et al., 2010*; *Lynch et al., 2016*; *Picardo et al., 2016*).

## Peri-Song activity in populations of $HVC_{RA}$ neurons

The sequence of syllables in zebra finch song is stereotyped and unfolds in less than a second. Like other rapid and precise motor movements, song may benefit from motor planning and preparatory activity unfolding on much longer timescales than the synaptic delays associated with descending motor commands, which in zebra finches are estimated to be ~25–50 ms (*Amador et al., 2013*; *Fee et al., 2004*); however, $HVC_{RA}$ neurons have been hypothesized to represent temporal sequences for songs and calls, and to remain quiescent at other times (*Hahnloser et al., 2002*; *Hamaguchi et al., 2016*; *Kozhevnikov and Fee, 2007*; *Long et al., 2010*). We examined the activity of $HVC_{RA}$ neurons prior to song onset, in between song bouts, and immediately after singing (*Figure 2a–c*, *Figure 2—figure supplement 1*). Song onset was defined by the onset of introductory notes that preceeded the bird's song phrase. A song phrase was defined as one or more repetitions

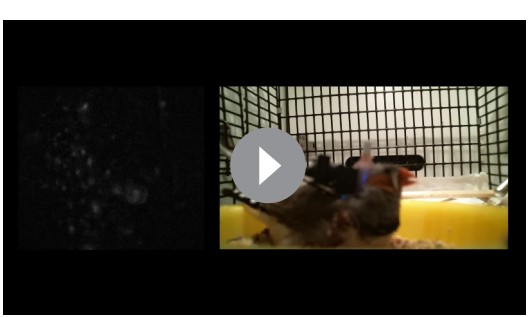

**Video 1.** Synchronized video of calcium imaging and behavior in a bird singing to a female.

DOI: https://doi.org/10.7554/eLife.43732.005

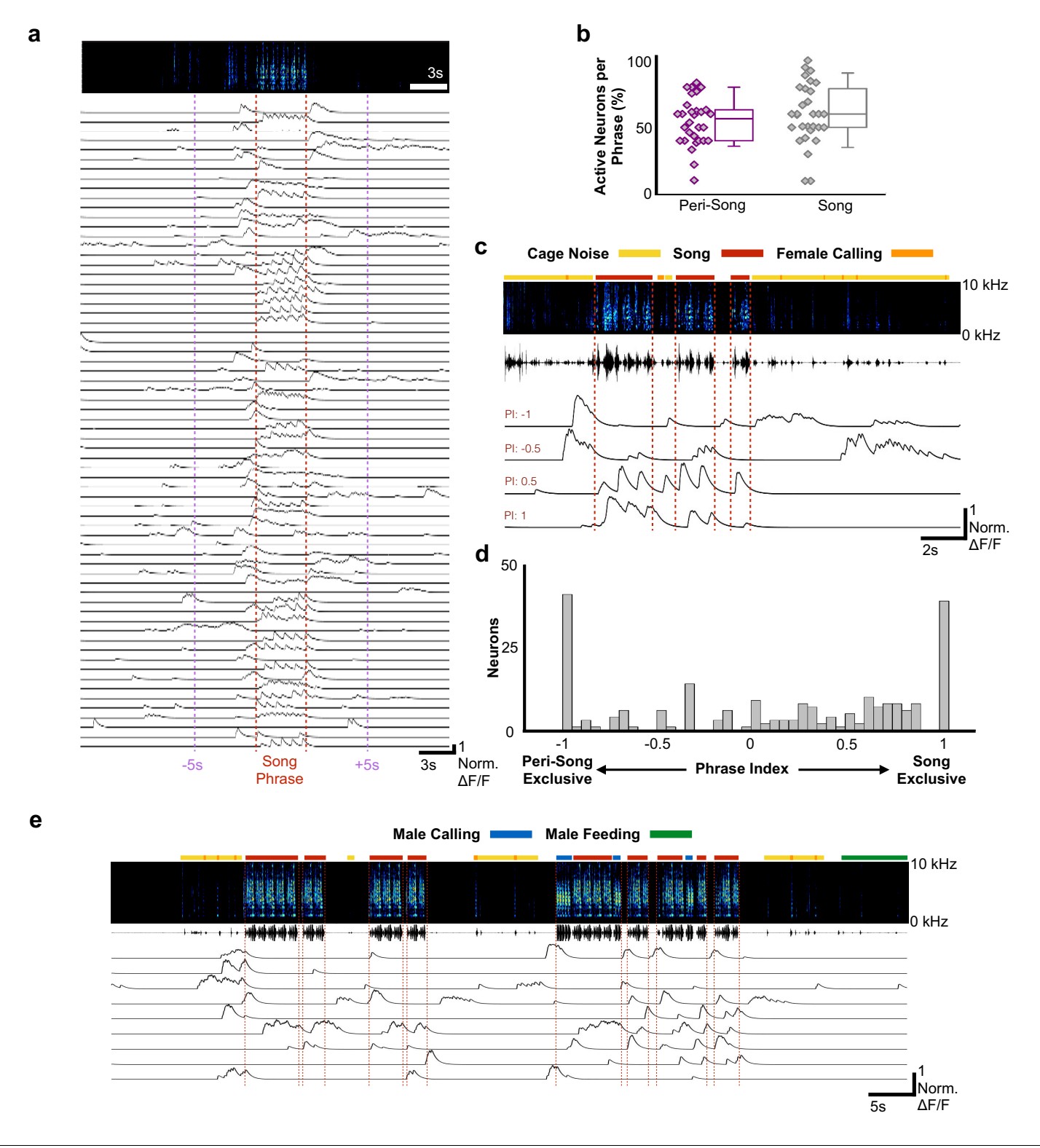

**Figure 2.** Most HVC_RA neurons exhibit peri-song activity. (a) Normalized calcium transients from 67 simultaneously recorded HVC_RA neurons during production of a song phrase. The red dashed lines delimit five consecutive motifs. (b) Percentages of active neurons during peri-song (54.3 ± 17.8%, SD, purple) and song (59.9 ± 22.5%, gray) are similar (30 phrases, $t = -1.1$, p=0.29 paired two-sample $t$ test). Box plots show the median, 25th and 75th percentiles with whiskers showing ±1.5 interquartile range (IQR). (c) Sample neurons with diverse phrase indices ranging from −1 to 1 and their corresponding calcium traces during 6 motifs over three bouts. Dashed lines indicate bout onsets and offsets. Bars above spectrogram indicate the

*Figure 2 continued on next page*

*Figure 2 continued*

presence of cage noise related to hopping and wing flapping (yellow) or female calling (FC, orange). (d) Histogram of phrase indices for all 223 neurons from six birds. (e) Undirected song from a different male showing periods of cage noise or hopping behavior (yellow) and feeding behavior (green). Blue boxes indicate the male calling. Red dashed lines indicate onsets and offsets of song bouts.

DOI: https://doi.org/10.7554/eLife.43732.006

The following source data and figure supplements are available for figure 2:

**Source data 1.** Raw active neuron numbers for *Figure 2B*.

DOI: https://doi.org/10.7554/eLife.43732.018

**Source data 2.** Raw neuron phrase index values for *Figure 2D*.

DOI: https://doi.org/10.7554/eLife.43732.019

**Figure supplement 1.** Description of motifs, bouts, and phrases underlying zebra finch courtship song structure.

DOI: https://doi.org/10.7554/eLife.43732.007

**Figure supplement 2.** Proportion of imaged neurons by bird (N = 6 birds, 197 neurons) that exhibited calcium events only before song onset calcium events (Pre-only, 0.44 ± 0.31, SD), only after song offset calcium events (Post-only, 0.26 ± 0.28), or were active before and after song (Pre-Post, 0.29 ± 0.25).

DOI: https://doi.org/10.7554/eLife.43732.008

**Figure supplement 3.** 62 events (31 paired pre-song and post-song events) from 31 neurons from six birds.

DOI: https://doi.org/10.7554/eLife.43732.009

**Figure supplement 4.** Spatial organization of HVC-RA neuron activity in one exemplary bird.

DOI: https://doi.org/10.7554/eLife.43732.010

**Figure supplement 5.** Pan-song neuron events organized in tripartite groups according to phrase index.

DOI: https://doi.org/10.7554/eLife.43732.011

**Figure supplement 6.** Synchronized calcium traces for all available trials for one bird (O248) across directed, undirected, and non-singing behaviors.

DOI: https://doi.org/10.7554/eLife.43732.012

**Figure supplement 7.** Expanded spectrograms from *Figure 2*, showing female distance calls and tet calls during pre-song and post-song periods.

DOI: https://doi.org/10.7554/eLife.43732.013

**Figure supplement 8.** Expanded spectrograms from two examples in *Figure 2—figure supplement 6*, showing the spectral structure of cage noise during pre-song behavioral epochs.

DOI: https://doi.org/10.7554/eLife.43732.014

**Figure supplement 9.** Event rates for a bird during Directed (561 CEs, 1.67 ± 0.74 std), Undirected (252 CEs, 1.06 ± 0.66), and Non-Singing (53 CEs, 0.19 ± 0.27) periods of behavior (Directed song is significantly different from non-singing, F(2,10) = 4.83, p<0.05).

DOI: https://doi.org/10.7554/eLife.43732.015

**Figure supplement 10.** Comparison between maximum pre-song event rate to the number of introductory notes prior to motif onset across five birds (Mean Event Rate = 0.91 E/s, Mean Introductory notes =~2).

DOI: https://doi.org/10.7554/eLife.43732.016

**Figure supplement 11.** Event rate calculated across the full length of 5 trials for one bird (O262).

DOI: https://doi.org/10.7554/eLife.43732.017

of a bird's motif with less than 2 s between onset of the next song bout (see Materials and methods for detailed definitions of phrases, song, and peri-song behavior). For any given song phrase, we observed significant activity surrounding singing behavior, indicating that $HVC_{RA}$ neuronal activity is not restricted only to the production of precise neural sequences for song in waking birds. 30 out of 30 song phrases from six birds exhibited 'peri-song' activity, defined as the 5 s intervals before and after singing, plus silent gaps within song phrases. Slightly fewer $HVC_{RA}$ neurons were active during peri-song intervals (54.3% of neurons) than during singing (59.9%), ($t = -1.1$, p=0.29 two-sample $t$ test; *Figure 2b*). Of the neurons that were active during peri-song intervals, 40.6% were active prior to song onset, 17.8% were active following song offset, and 41.6% were active before and after song (N = 197 neurons, six birds, *Figure 2—figure supplement 2*). When we examined the timing of peri-song events, we found no correlation in the timing of pre-song and post-song event onsets in neurons that were sparsely active both before and after song (*Figure 2—figure supplement 3*). Although neuronal populations displayed considerably more calcium events during song ($t = 5.635$, p=$5.65 \times 10^{-7}$ two-sample $t$, normalized song event rate = 24.09 events/s±15.95 SD, normalized peri-song event rate = 6.97 events/s±3.72 SD), a substantial fraction of all recorded calcium events occurred within the 5 s intervals before or after song (997/2366 or 32.5% of all calcium events).

To better characterize the activity profiles of $HVC_{RA}$ neurons, we indexed the song and peri-song activity of all $HVC_{RA}$ neurons throughout a day of singing (phrase index: range −1 to +1, with

neurons exclusively active outside of singing scoring −1 and neurons active only during singing +1, *Figure 2c–d*, see Materials and methods). We found that $HVC_{RA}$ phrase indices were not uniformly distributed ($\chi^2$ (7, N = 223)=46.3, p=7.6×10$^{-8}$, Chi-square goodness of fit test), with a significant fraction (36%) falling at the extremes of this scale (*Figure 2d*), and that neurons with different phrase indices were anatomically intermingled throughout HVC (*Figure 2—figure supplement 4*). Most neurons (64.1%) displayed sparse heterogenous activity during peri-song periods and transitioned to temporally precise activity during singing (referred to here as 'pan-song neurons'; *Figure 2—figure supplement 5*; *Supplementary file 2*). At the extremes of the phrase index scale (phrase indices −1 or +1), we found that 18.4% of neurons were exclusively active during peri-song intervals (phrase index = −1, referred to here as 'peri-song neurons'; *Supplementary file 2*), while 17.5% participated exclusively in neural sequences during singing (phrase index =+1, referred to here as 'song neurons'; *Supplementary file 2*).

These results reveal that a substantial portion of $HVC_{RA}$ neurons are active outside of the precise neuronal sequences associated with song, expanding our view of the potential functional role of this neuronal population. That more than half of all $HVC_{RA}$ neurons can be active during peri-song intervals raises the prospect that precise neural sequences emerge as part of changing network dynamics across subpopulations of $HVC_{RA}$ neurons. These results may also lend insight into why approximately half of $HVC_{RA}$ neurons recorded using electrophysiological methods appear to be inactive during song (*Fee et al., 2004*; *Hamaguchi et al., 2016*; *Kozhevnikov and Fee, 2007*; *Long et al., 2010*). Previous multi-unit recordings in zebra finches and mockingbirds, which are dominated by activity of interneurons or neurons projecting to the basal ganglia, have identified 'anticipatory' activity in HVC hundreds of milliseconds prior to song onset, but the role of this activity and whether $HVC_{RA}$ neurons are active prior to song onset have not been examined (*McCasland, 1987*; *Rajan, 2018*; *Rajan and Doupe, 2013*).

These results also raise questions as to why previous studies in adult zebra finches have not identified peri-song activity in $HVC_{RA}$ neurons. Although speculative, several reasons may account for this. First, previous calcium imaging experiments have not restricted GCaMP expression to $HVC_{RA}$ neurons, but rather relied on either non-selective labeling of neuronal populations in HVC (*Katlowitz et al., 2018*; *Liberti et al., 2016*; *Markowitz et al., 2015*; *Picardo et al., 2016*) or have been restricted to imaging small populations of other classes of HVC neurons (*Roberts et al., 2017*). Second, electrophysiological studies of identified $HVC_{RA}$ neurons have been mostly confined to recording one neuron at a time. These experiments have focused on understanding coding during song production and at least initial studies used short (~500 ms) buffering windows triggered by singing behavior (*Hahnloser et al., 2002*). Therefore, sparse heterogenous activity occurring seconds before or after song could be simply overlooked or could appear irrelevant unless viewed through the lens of population dynamics. Third, it is possible that there are incongruencies between spiking activity and calcium signals, as has been shown in the feed-forward inhibitory control of bursting activity in the striatum (*Owen et al., 2018*). Although several studies have examined the relationship between calcium responses and suprathreshold activity in HVC projection neurons (*Graber et al., 2013*; *Peh et al., 2015*; *Picardo et al., 2016*), this relationship has not been tested during peri-song periods in freely singing birds.

It is also possible that peri-song activity is unrelated to singing and merely reflects low levels of spontaneous activity intrinsic to $HVC_{RA}$ neurons. This was not the case, however, as $HVC_{RA}$ neurons were largely inactive outside of peri-song intervals and were significantly more active during the peri-song periods than baseline (baseline calculated from periods $\geq$ 10 s removed from periods of singing or calling, p=8.4×10$^{-5}$, Chi-square = 18.78 Friedman test, baseline = 12.9% ± 5.7 SD of fluorescence values normalized to song, pre-song = 26.9% ± 14.7, and post-song = 30.2% ± 10.7). In addition, we examined the amplitudes of peri-song calcium events and found that they were larger than events occurring during song ($t$ = 3.2769, p=0.0012, two-tailed $t$ test, 279 fluorescence peaks measured from neurons active during both peri-song and song, pan-song neurons with phrase indices between −0.18 to 0.18). We also asked whether peri-song activity might relate to factors other than singing. We examined trials in which birds did not sing to female birds but did not find $HVC_{RA}$ neurons that responded solely to presentation of the female or during non-song related movements of the head, beak, or throat, such as during eating, grooming, and seed-shelling (*Figure 2e, Figure 2—figure supplement 6, Figure 2—figure supplement 7, Figure 2—figure supplement 8*). Indeed, populations of $HVC_{RA}$ neurons only became substantially active prior to singing or calling.

Moreover, in a single bird in which we were able to image neuronal activity during undirected singing, that is song produced when the bird was alone in its cage, we also found peri-song activity in the moments before and after (*Figure 2—figure supplements 6–9*), suggesting that peri-song activity is unlikely to be solely associated with extraneous nonvocal singing behaviors such as courtship dance.

We next examined the possibility that $HVC_{RA}$ neurons play a role in motor planning or preparation as birds prepare to sing. We found pre-song activity in 28/30 song phrases analyzed. In the two instances when we did not detect any pre-song activity, less than 6 $HVC_{RA}$ neurons were active within our imaging window during singing, indicating that the lack of activity was likely the result of under sampling from the population. Electrophysiological recordings in young zebra finches have identified HVC neurons that mark the onset of song bouts (*Okubo et al., 2015*), 'bout neurons' that burst immediately prior to vocalizations. The vast majority of pre-song activity we describe occurs hundreds of milliseconds to seconds prior to vocalizations, suggesting a role in planning or preparation to vocalize (96.2% of calcium events occurred more than 100 ms prior to vocal onset and 65.6% occurred more than 1 s prior to vocal onset). Zebra finches often sing a variable number of introductory notes prior to the first motif of a song bout. Pre-song activity (prior to introductory notes) could be related to the number of introductory notes to be sung, but we found no correlation between pre-song event rates and the number of introductory notes (*Figure 2—figure supplement 10*). In 27/28 song phrases we found increases in population activity greater than 3 SDs above baseline predicted song onset within the following 4–5 s (2.44 s ± 1.0 s SD, 28 song phrases from five birds). Calcium activity reached two-thirds of the maximum pre-song activity only prior to song onset or prior to short vocalizations (*Figure 2—figure supplement 11*). Together, these results indicate that $HVC_{RA}$ neuron activity is predictive of the voluntary production of courtship song and suggests a role for this network in motor planning and in preparation to sing.

## Peri-Song and Pan-Song neurons

A substantial fraction of all imaged neurons (41/223 neurons) were active exclusively before or after singing (*Figure 2d*, neurons with a phrase index of −1). These peri-song neurons exhibited sparse heterogeneous activity before song phrases and were occasionally active in the silent intervals between song motifs (*Figure 3a–c*, and *Figure 3—figure supplement 1*). Both the number of active peri-song neurons and the density of calcium events increased 1–3 s prior to song onset, with the event rate peaking 1.5 s prior to singing (*Figure 3b–c,i*) and then declining sharply in the last second before song onset (*Figure 3i*). Most of the neurons we imaged, pan-song neurons (143/223 neurons), exhibited sparse, heterogeneous activity before and/or after song and exhibited time-locked sequences during singing (*Figure 3d–f*). Pan-song neurons exhibited substantial increases in their activity in the last ~2 s prior to song onset. Their activity continued to increase as the activity of peri-song neurons began to ramp-off prior to song onset (*Figure 3g,l*, and *Figure 3—figure supplement 2*). Differences in pre-song activity profiles between peri-song and pan-song neurons (Kolmogorov-Smirnov, K-S test, $k = 0.26$, p=0.056) may support temporally coordinated network transitions as birds prepare to sing, suggesting that neural sequences for song could emerge as part of changing network dynamics in HVC.

During production of song motifs, pan-song neurons exhibited sequential bursts of activity that roughly coded for all moments in the bird's song, similar to sequencing previously described in song neurons. We found that pan-song and song neurons had a similar probability of being active within each motif (song neurons probability of at least one calcium event per motif P(motif)=0.72 ± 0.32, pan-song neurons P(motif)=0.66 ± 0.29, K-S test, p=0.11; *Figure 3—figure supplement 3*); however, we also noted that the probability of being active was lower than in electrophysiological recordings (*Hahnloser et al., 2002*; *Kozhevnikov and Fee, 2007*). This likely reflects limitations in event detection using single-photon calcium imaging. To better understand this, we calculated signal to noise ratios (SNRs, signal defined as peak fluorescence of calcium events during song) between pan-song neurons and song neurons during singing (SNR song neurons = 913.8 ± 405.4 (7 neurons), pan-song neurons = 638.3 ± 248.2 SEM, n = 36 neurons). We found no difference in SNR between these neurons (two-sample t test, p=0.6; SNR calculated from 156 calcium events (28 from song neurons and 128 from pan-song neurons), suggesting that although we are underestimating activity during singing, these limitations are unlikely to obscure differences between pan-song and song neurons.

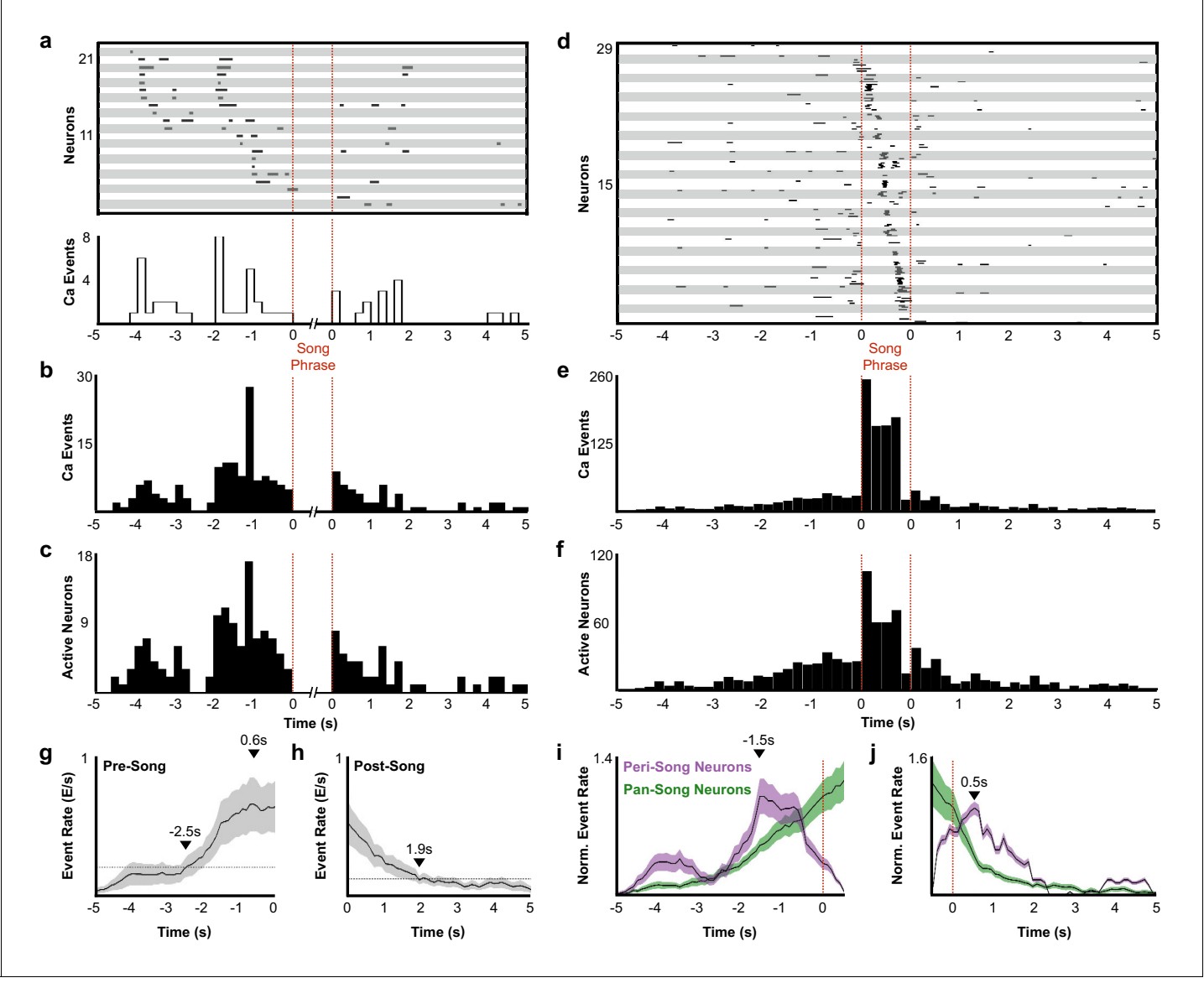

**Figure 3.** Description of peri-song and pan-song neuron activity. (**a**) 21 peri-song neurons from one bird singing 3 bouts containing six motifs (1 st bout: three motifs; 2nd bout: two motifs; 3rd bout: one motif, the dashed red lines indicate the onset and offset of the three bouts). Each row shows song-aligned calcium events (CEs N = 54 CEs; average rise time = 0.18 ± 0.09 s SD). The shaded horizontal bars separate different neurons. One CE is seen to overlap with the beginning of the song phrase. The onset time for this event is 170 ms before song, but the rise time is slow and extends to 100 ms after song onset. Below the CE raster plot is a peri-event histogram with the event rate in 200 ms bins shown for the trial above. (**b**) Song-aligned CEs in peri-song neurons 5 s before and after phrase onset (41 neurons, 190 CEs). The activity rate peaks ~ 1.2 s before phrase onset. (**c**) The number of active peri-song neurons in 200 ms bins before and after phrase onset (41 neurons, 169 CEs). (**d**) Song-aligned activity of pan-song neurons (same trial as shown in panel a), N = 29 neurons, 253 CEs. (**e**) Peri-event histogram of pan-song neurons (143 neurons, 1,333 CEs). (**f**) The number of active pan-song neurons in 200 ms bins (143 neurons, 853 CEs). (**g**) Pre-song event rate for all neurons. The event rate was calculated by counting event onsets in 100 ms bins and then smoothed with a 1 s moving window. 28 trials are shown from five birds; the black line indicates the average event rate. The black triangles mark the peak event rate occurring 0.6 s before song onset and 2.5 s when the event rate reaches 3 SD above the baseline event rate, respectively. Baseline event rate was determined by measuring the average event rate from −5 to −4 s before song onset. Shaded region indicates standard deviation. (**h**) Post-song event rate for all neurons. 27 trials are shown from five birds. The black triangles mark when the event rate reaches 3 SD above the baseline event rate. Baseline event rate was determined by measuring the average event rate during −5 to −4 s before song onset. (**i**) Pre-song event rate for peri-song and pan-song neurons calculated as calcium events in moving 1 s windows. The black line indicates average event rate. The black triangle indicates peak event rate occurring 1.5 s before phrase onset. (**j**) Same as i), but post-song event rates for peri-song and pan-song neurons. The black triangle indicates peak event rate occurring 0.5 s after phrase offset.

DOI: https://doi.org/10.7554/eLife.43732.020

The following source data and figure supplements are available for figure 3:

*Figure 3 continued*

**Source data 1.** Raw pre-song event rates for *Figure 3G*.
DOI: https://doi.org/10.7554/eLife.43732.024
**Source data 2.** Raw post-song event rates for *Figure 3H*.
DOI: https://doi.org/10.7554/eLife.43732.025
**Source data 3.** Raw pre-song event rates for peri-song and pan-song neurons in *Figure 3I*.
DOI: https://doi.org/10.7554/eLife.43732.026
**Source data 4.** Raw post-song event rates for peri-song and pan-song neurons in *Figure 3J*.
DOI: https://doi.org/10.7554/eLife.43732.027
**Figure supplement 1.** Inter-bout events for peri-song and pan-song neurons.
DOI: https://doi.org/10.7554/eLife.43732.021
**Figure supplement 2.** Distribution of calcium events and active neurons based on neuron category.
DOI: https://doi.org/10.7554/eLife.43732.022
**Figure supplement 3.** Comparison of the probability of pan-song neuron and song neuron events occurring during a motif (Kolmogorov-Smirnov test, n.s.
DOI: https://doi.org/10.7554/eLife.43732.023

In addition to preceding song onset, neurons also marked the end of song phrases. Within 5 s after song offset, peri-song neurons exhibited a sharp increase in activity followed by a gradual ramp-off (*Figure 3a–c,h,j*, and *Figure 3—figure supplement 2*) whereas pan-song neurons only exhibited a ramp-off (*Figure 3d–f,h,j*). Although the distribution of post-song activity differed between peri-song and pan-song neurons (K-S test, $k$ = 0.2692, p=0.0373), both populations returned to baseline activity over similar timescales. The function of post-song activity is unclear but may provide a circuit mechanism for birds to rapidly re-engage in song performances given appropriate social feedback or context. Courtship singing is tightly coupled to social interaction with female birds and it is common for male birds to string two or more song phrases together during courtship song (*Williams, 2004*) (see *Figure 2—figure supplements 1* and *6*). Post-song activity could also reflect moments when birds are unable to continue singing due to hyperventilation induced by the rapid respiratory patterns associated with song (*Franz and Goller, 2003*). To explore this idea, we examined whether the duration of song phrases was correlated with the number of active neurons in the post-song period, but did not find a significant correlation ($r^2$ = 0.03). Although the function of post-song activity is unclear, our results indicate that pre-song activity forecasts impending song and suggest a previously unappreciated role for the HVC$_{RA}$ network in planning or preparing to sing.

## Common preparatory activity in premotor circuits across multiple species

To examine whether preparatory activity is a common circuit mechanism for the production of bird-song, we recorded HVC and RA neural activity in another songbird species, Bengalese finches (*Lonchura striata domestica*). Motor planning and preparation facilitate the accurate execution of fast and precise movements, which are common to the songs of zebra finches and Bengalese finches, however, syllable sequences in Bengalese finches are less stereotyped than those in zebra finches (*Okanoya, 2004*). Using multi-channel neural recordings in HVC, we identified robust preparatory activity several hundreds of milliseconds prior to song onset (*Figure 4a–g*). The pre-song and song–related multiunit spike rates were significantly above baseline (Wilcoxon signed-rank test after Bonferroni correction, n = 29 MU sites. pre-song: $z$ = 4.62, p=0.030; song: $z$ = 4.70, p<0.001), whereas the post-song spike rates was not ($z$ = 2.17, p=0.089). Pre-song activity increased above baseline $-1.47 \pm 1.05$ s prior to song onset, a timescale that closely matched the timing of peak calcium-event rates in peri-song neurons in zebra finches. The offset timing of post-song activity was $0.42 \pm 0.23$ s. This indicates that preparatory activity in HVC is a common network motif important for song generation and the onset of precise neural sequences.

HVC contains multiple cell types, including interneurons and at least three different classes of projection neurons (*Mooney and Prather, 2005*; *Roberts et al., 2017*). Multichannel recordings in HVC provide an important read-out of the network activity prior to song onset but alone are insufficient to assess whether this preparatory activity influences descending cortical pathways involved in song

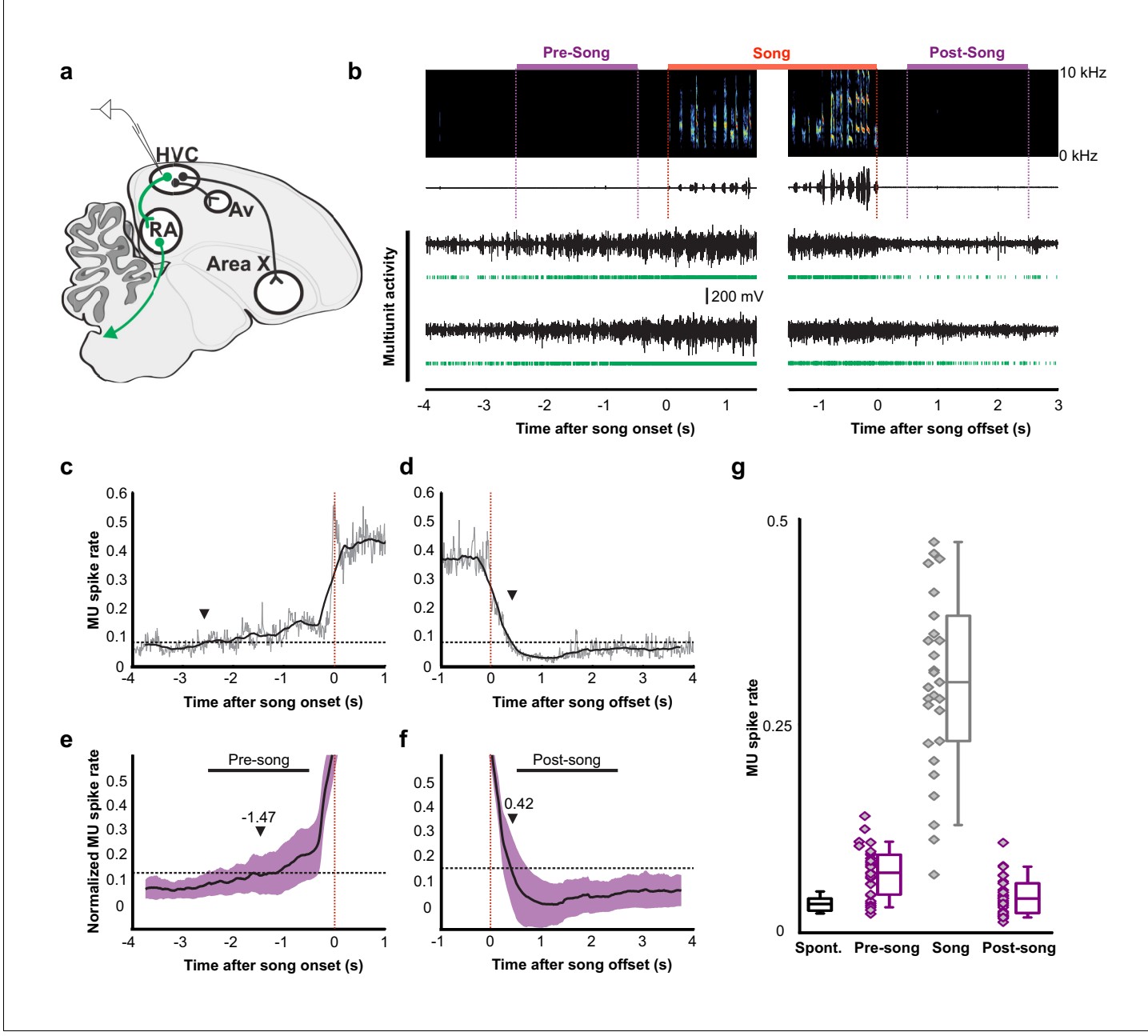

**Figure 4.** Pre-song and post-song firing in HVC of Bengalese finches. (a) Schematic of recording site. (b) An example of song initiation and termination (dashed lines indicate phrase onset and offset, pre and post-song period marked in purple and song marked in red) and simultaneously recorded HVC multiunit activity on two electrodes (channels #4 and #15, bird p15o56). Green raster plots represent detected spikes on the two electrodes. (c,d) Phrase onset (c) and offset (d) related average multiunit activity obtained from an example electrode channel (vertical dashed lines indicate phrase onset and offset, respectively). The spike rate was averaged across multiple song onsets or offsets and was first calculated in 10 ms bins (gray thin line) and then smoothed with a 500 ms window (bold line). Upper and lower horizontal dotted lines show mean spike rates during singing and baseline, respectively. Arrowheads indicate onset (c) and offset (d) timings of spike rate, as assessed by crossing of a pre-defined threshold (red line). (e,f) Normalized multiunit activity related to phrase onset (e) and offset (f). Before averaging, the spike rate trace of each electrode channel was normalized such that 0 corresponds to the mean rate during baseline and 1.0 to the mean rate during singing (see Materials and method). The bold line shows an average across all electrodes and birds (n = 29 channels). Purple area indicates ±1.0 SD. Arrowheads show mean onset (e) and offset (f) timing of pre-song and post-song activity, respectively. (g) Mean multiunit spike rates during spontaneous (black), pre-song (purple with diamonds), song (gray with diamonds), and post-song (purple with diamonds) periods. Pre-song and post-song periods are indicated by the horizontal bars in panels e and f. Box plots show the median, 25th and 75th percentiles with whiskers showing ±1.5 IQR.

DOI: https://doi.org/10.7554/eLife.43732.028

*Figure 4 continued on next page*

*Figure 4 continued*

The following source data is available for figure 4:

**Source data 1.** Raw multiunit spike rate averages for *Figure 4G*.
DOI: https://doi.org/10.7554/eLife.43732.029

motor control. Therefore, we recorded single unit electrophysiological activity from the downstream targets of $HVC_{RA}$ neurons within the cortical premotor nucleus RA. Projection neurons in RA are tonically active at baseline (*Figure 5a*, 'Spontaneous') and exhibit precise bursts of activity during singing (*Figure 5a*, 'Song'), a transition well captured by changes in the coefficient of variation of the inter-spike intervals ($CV_{ISI}$, *Figure 5c*). We measured changes in RA neuron activity in the period just prior to song initiation and just after the conclusion of each song bout (between 0.5 and 2.5 s before/after the first/last song syllable). As expected, we did not find substantial differences in spike rates between non-singing and singing states (not shown) but found that the $CV_{ISI}$ for RA neurons changed significantly during song, reaching higher values during pre-song, song, and post-song epochs as compared to spontaneous activity (*Figure 5d* and *Figure 5—figure supplement 1*, p<0.005, two-sided K-S tests). This suggests that $HVC_{RA}$ sequences associated with preparation to sing propagate to downstream premotor circuits prior to song onset and that HVC continues to influence descending motor pathways following song cessation.

## Peripheral preparation to sing

Pre-song activity could reflect motor planning (changes in network activity independent of changes in the motor periphery) and/or motor preparation that functions to coordinate changes in the motor periphery as birds prepare to sing. Song is a respiratory behavior that is primarily produced during expiration and silent intervals in the song correspond to mini-breaths, which are rapid, deep inspirations (*Hartley and Suthers, 1989*; *Schmidt and Goller, 2016*). How birds plan to sing or prepare the respiratory system to sing is poorly understood, but there is evidence that prior to song onset, oxygen consumption decreases and respiratory rate increases (*Franz and Goller, 2003*). To explore the time course of changes in respiratory patterns in more detail, we used air sac pressure recordings in singing zebra finches (*Figure 6a–c*). During singing, birds significantly accelerated the respiratory rhythm and marginally shifted towards longer periods of expiration during each cycle (*Figure 6b*, respiratory duration pre-song = 0.38 ± 0.04 s (SD), song = 0.18 ± 0.03 s, post-song = 0.39 ± 0.06 s: $F(2,10)$ = 46.63, p<0.001; duty cycle (% of time in expiration) pre-song = 58% ± 3 (SEM), song = 57% ± 2.5, post-song = 61% ± 2.4: $F(2,10)$ = 3.56, p=0.07). We found that significant changes in respiration also preceded song onset. Respiratory cycle duration significantly accelerated in the last second prior to song onset with relative decreases of expiratory phases (respiratory duration: $F(3,15)$ = 7.67, p=0.02, duty cycle: $F(3,15)$ = 4.077, p=0.07; *Figure 6d*). Following song termination, birds immediately returned to longer respiratory cycles but during the first second post-song, they spent more time exhaling compared to inhaling, a behavior likely involved in helping to recover from singing-related hyperventilation (respiratory duration: $F(2,10)$ = 0.509, p<n . s., duty cycle: $F(2,10)$ = 6.553, p<0. 01; *Figure 6e*). These changes in respiration during the last second before and first second following song support the idea that $HVC_{RA}$ neurons provide descending motor commands that coordinate transitions between non-vocal and vocal states by coordinating respiratory patterns. The lack of changes at earlier time-points prior to singing also indicate that pre-song activity 1–3 s prior to song onset may reflect motor planning or the decision to sing, rather than respiratory preparation. Together these, findings support the idea that $HVC_{RA}$ neurons could function in aspects of motor planning as well as preparation.

## Discussion

Previous studies suggested that the $HVC_{RA}$ network functions as a time-keeper, encoding motif-level temporal representations of song via propagation of precisely timed neural sequences (*Hahnloser et al., 2002*; *Kozhevnikov and Fee, 2007*; *Long and Fee, 2008*; *Long et al., 2010*; *Lynch et al., 2016*; *Markowitz et al., 2015*; *Picardo et al., 2016*). Central to this view is that $HVC_{RA}$ neurons are active during singing and hence behave in primarily two modes, inactive or

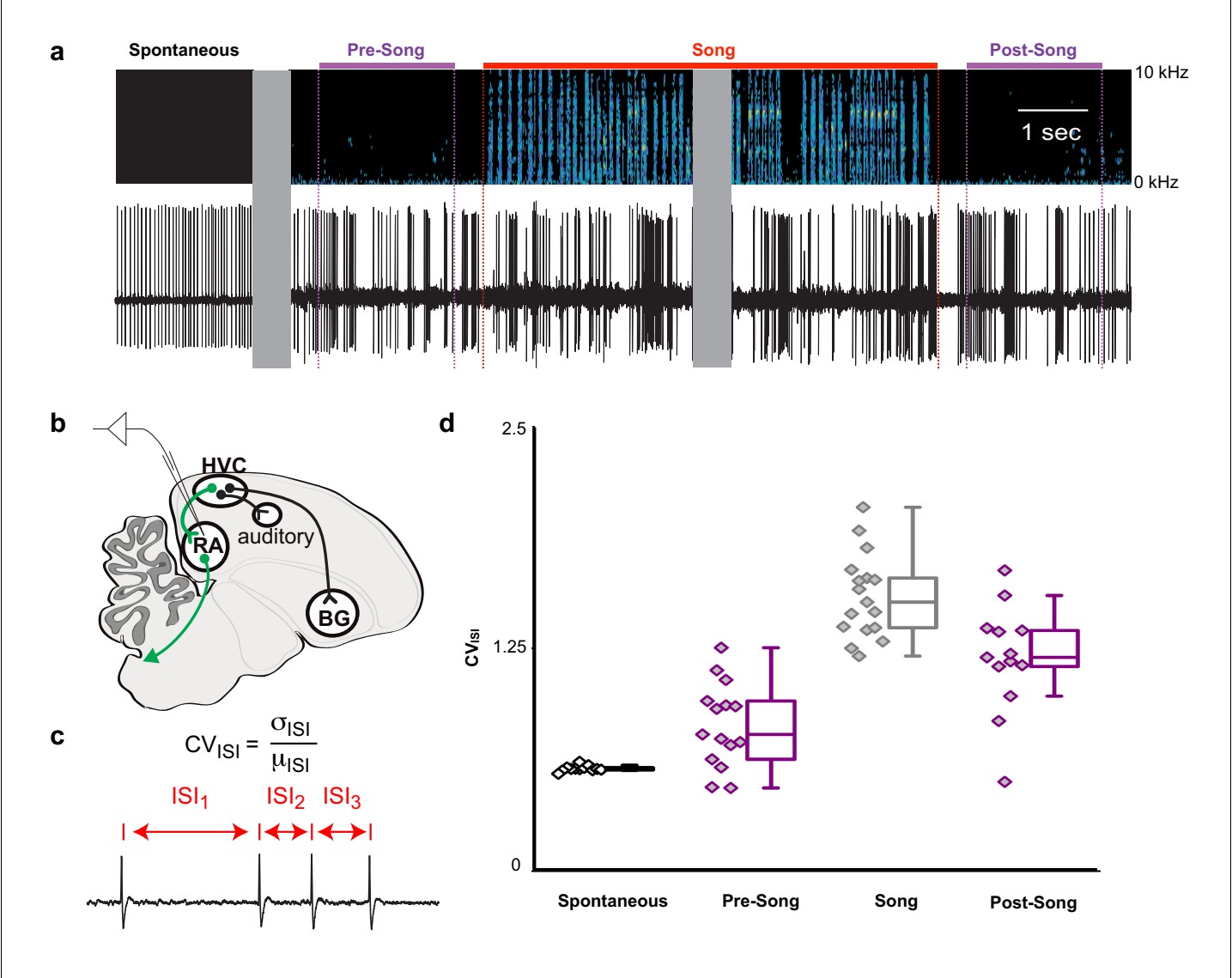

**Figure 5.** Pre-song and post-song firing in RA of Bengalese finches. (**a**) Example extracellular recording from a single RA neuron. Colored lines highlight four epochs (pre-song (purple), song (red), and post-song (purple)) relative to the beginning and ending of a song phrase (see main text). Gray areas indicate discontinuities in time (pauses between 'spontaneous' epoch and song initiation and within the middle portion of the song bout). (**b**) Schematic of recording site. (**c**) We quantified inter-spike-intervals (ISIs) and computed the coefficient of variation (CV) in each epoch. (**d**) We found significantly higher ISI variability in the pre-song epoch (purple with diamonds) compared to spontaneous (p<0.005, two-sided K-S test). Box plots show the median, 25th and 75th percentiles with whiskers showing ±1.5 IQR.

DOI: https://doi.org/10.7554/eLife.43732.030

The following source data and figure supplement are available for figure 5:

**Source data 1.** Raw coefficient of variation data for *Figure 5D*.
DOI: https://doi.org/10.7554/eLife.43732.032

**Figure supplement 1.** Pre-song changes in $CV_{ISI}$ in 600 ms bins from single units in the RA of Bengalese finches singing undirected song.
DOI: https://doi.org/10.7554/eLife.43732.031

propagating neural sequences. Our principal result is that neural sequences among $HVC_{RA}$ neurons emerge as part of orchestrated population activity across a larger network of $HVC_{RA}$ neurons. This activity can be correlated with motor planning and preparation prior to song initiation. Peri-song and pan-song $HVC_{RA}$ neurons forecast the start of singing. Peri-song $HVC_{RA}$ neurons are inactive during song, whereas pan-song neurons become heterogeneously active prior to time-locked

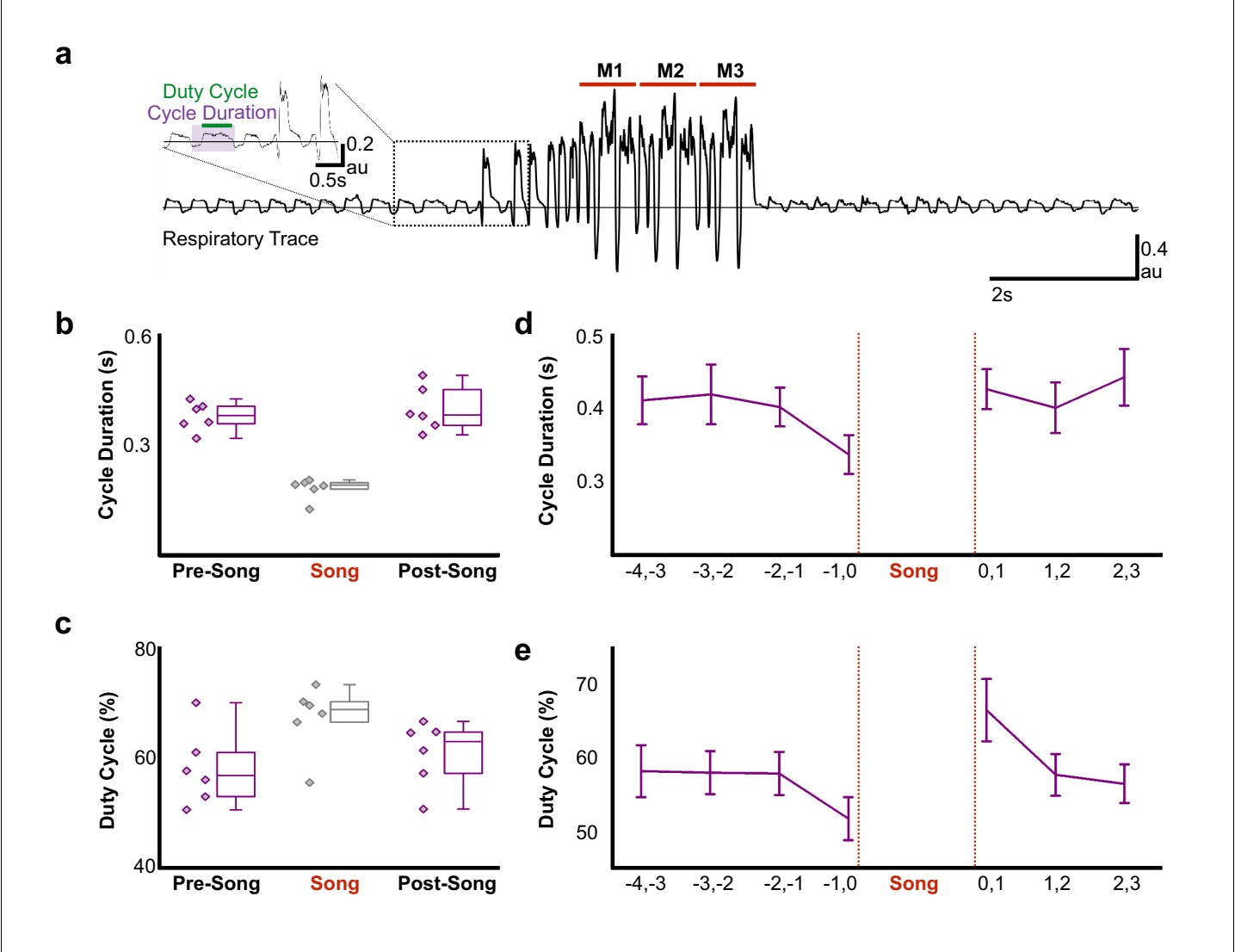

**Figure 6.** Air sac pressure recording in zebra finches. (a) Waveform of pressure changes during non-singing and singing periods. Waveforms above the horizontal line (suprambient pressurization) indicate expiration and below the line (subatmospheric pressurization) indicate inhalation. Song start was identified by the presence of introductory notes preceeding the song phrase. M1-3 corresponds to three repetitions of the bird's motif. Inset illustrates measurements for respiratory cycle duration and duty cycle (% time in expiration) and the first two introductory notes. (b) Respiratory cycle duration and (c) duty cycle of expiratory phase before (Pre), during (Song), and after (Post) song production (N = 6 birds). (d) Plots of average respiratory cycle durations and (e) duty cycles during pre-song and post song periods (N = 6 birds). Longer duty cycles correspond to increased periods of expiration. Data in panels b-e is derived from the same six birds.

DOI: https://doi.org/10.7554/eLife.43732.033

The following source data is available for figure 6:

**Source data 1.** Raw cycle duration and duty cycle values for *Figure 6B and 6C*.
DOI: https://doi.org/10.7554/eLife.43732.034
**Source data 2.** Raw cycle duration and duty cycle values binned by 1s time windows for *Figure 6D and 6E*.
DOI: https://doi.org/10.7554/eLife.43732.035

sequential activity during song performances (*Figure 2 and 3*). Moreover, we find that preparatory activity in HVC$_{RA}$ neurons precedes the pre-bout activity described in other classes of HVC neurons and other portions of the song system previously described in zebra finches (*Danish et al., 2017*; *Goldberg et al., 2010*; *Goldberg and Fee, 2012*; *Kao et al., 2008*; *Rajan, 2018*; *Roberts et al., 2017*; *Vyssotski et al., 2016*; *Williams and Vicario, 1993*) and described here in Bengalese finches

(*Figure 4*). This suggests that HVC$_{RA}$ neurons may seed network wide changes among other classes of HVC neurons and the song system more generally.

Indeed, the rigid stereotypy of singing behavior enables comparisons from different levels of the nervous system and periphery. We find that preparatory activity in HVC$_{RA}$ neurons drives descending motor commands via RA and motor movements that set the stage for producing song (*Figure 5 and 6*). Because song is a self-initiated, volitional behavior, our findings further indicate that the HVC$_{RA}$ network either functions as a sensitive read-out of the decision to sing or as an integral factor in the decision itself. Finally, previous studies have shown that about half of HVC$_{RA}$ neurons are inactive during song (*Hahnloser et al., 2002*; *Hamaguchi et al., 2016*; *Kozhevnikov and Fee, 2007*; *Long et al., 2010*). We find that approximately half of the HVC$_{RA}$ neuronal network is active at peri-song intervals, perhaps accounting for previous recordings from neurons that are inactive during singing. We therefore propose that one important function of HVC$_{RA}$ neurons is to plan and prepare for the upcoming song performance.

Together, these findings support a simple model for song and neural sequence initiation. Preparatory activity in populations of HVC$_{RA}$ neurons drives descending motor commands via RA and its connections to the ventral respiratory group and syringeal motoneurons in the medulla (*Andalman et al., 2011*; *Goller and Cooper, 2004*; *Roberts et al., 2008*; *Sturdy et al., 2003*; *Suthers et al., 1999*; *Wild, 1993*). Given the sparsity of HVC$_{RA}$ neuron activity and the convergence of HVC$_{RA}$ input to RA, it is likely that only population activity, like that described here, is sufficient to drive bursting in RA. These motor signals increase respiratory rate, bringing it closer to the high rate needed to coordinate production of song syllables. Because the initiation of singing requires precise coordination between respiratory state and descending motor commands, we hypothesize that recurrent projections from the brainstem update activity in HVC and trigger the initiation of neural sequences once the periphery is readied for the first respiratory cycle for song (*Ashmore et al., 2005*; *Hamaguchi et al., 2016*; *Schmidt et al., 2012*). Circuitry related to recurrent projections into HVC increase their activity tens of milliseconds prior to song onset (*Danish et al., 2017*; *Vyssotski et al., 2016*; *Williams and Vicario, 1993*) and thus may provide final inputs that help mediate the transition from heterogenous preparatory activity to the precisely sequenced activity that underlies song.

Absent from this model is how activity in peri-song and pan-song neurons is first initiated seconds before song onset. HVC receives input from cholinergic neurons in the basal forebrain (*Li and Sakaguchi, 1997*; *Shea et al., 2010*; *Shea and Margoliash, 2003*), noradrenergic neurons in the locus coeruleus (*Appeltants et al., 2000*), and dopaminergic neurons in the midbrain (*Hamaguchi and Mooney, 2012*; *Tanaka et al., 2018*), any or all of which could play potent roles in shifting the excitability of subsets of HVC$_{RA}$ involved in song preparation and initiation. Neuromodulatory inputs may also play a role in shifting the excitability of the entire song system as birds prepare to sing. Further studies will be needed to understand the function of neuromodulatory inputs to HVC$_{RA}$ neurons and the rest of the song circuitry in song initiation.

Also absent from our model are specific predictions about the role peri-song neurons and pan-song neurons may play in motor planning. Although it is not yet known whether activity occuring among peri-song and pan-song HVC$_{RA}$ neurons prior to song onset is necessary for song initiation or production, previous studies have identified pyramidal tract neurons (PT$_{upper}$) in the anterior lateral motor cortex that play a specific role in motor planning (*Economo et al., 2018*). These neurons are active prior to movement onset and show decreased activity during movement, similar to the activity of HVC$_{RA}$ peri-song neurons described here. PT$_{upper}$ neurons innervate the thalamus rather than medulla and are hypothesized to encode cognitive signals related to motor planning (*Economo et al., 2018*). Likewise, RA provides inputs to the medulla and the thalamus (*Goldberg and Fee, 2012*; *Roberts et al., 2008*; *Vates et al., 1997*; *Wild, 1993*). However, it is currently not known if subpopulations of RA neurons project exclusively to the thalamus or if peri-song and pan-song HVC$_{RA}$ neurons have unique downstream targets within RA. Future experiments, requiring novel closed-loop manipulations to exclusively disrupt the activity of functionally identified peri-song or pan-song neurons, independent of activity associated with the motif-level temporal representations of song, and detailed circuit mapping studies will be needed to dissect the function of these newly identified subpopuations of HVC$_{RA}$ neurons. When possible, such experiments will undoubtedly lend insights into whether subpopulations of HVC$_{RA}$ neurons are involved in aspects of motor planning independent of motor preparation or song performance.

Neural activity associated with motor planning and preparation has been observed in motor and premotor cortices for a variety of different motor tasks in rodents and primates (*Chen et al., 2017*; *Churchland et al., 2010a*; *Churchland et al., 2006a*; *Churchland et al., 2006b*; *Economo et al., 2018*; *Inagaki et al., 2019*; *Kaufman et al., 2014*; *Li et al., 2015*; *Li et al., 2016*; *Murakami and Mainen, 2015*; *Murakami et al., 2014*; *Tanji and Evarts, 1976*), including vocalizations (*Gavrilov et al., 2017*). This activity is thought to reflect the decision to perform movements and is characterized by a high degree of variability from trial to trial. Changes in circuit dynamics function to shift the initial state of a network to levels that enable efficient and accurate motor performances and this activity can start to unfold seconds prior to movement initiation (*Churchland et al., 2010a*; *Inagaki et al., 2019*; *Murakami and Mainen, 2015*; *Svoboda and Li, 2018*). HVC is proposed to be analogous to the mammalian motor cortex (layer III neurons of the primary motor cortex) (*Pfenning et al., 2014*), or premotor cortex (*Bolhuis et al., 2010*). Our observation of preparatory activity seconds before the onset of courtship song in two different songbird species suggests that pre-movement activity is a common mechanism for ensuring the accurate production of volitional behaviors. In line with recordings of pre-motor activity in mammals, individual $HVC_{RA}$ neurons exhibit a high degree of trial to trial variability. Although zebra finch courtship song is famously stereotyped, there is a measurable degree of variability in the structure and duration of song motifs, bouts and phrases from trial to trial. Recording of preparatory activity across larger populations of $HVC_{RA}$ neurons may be useful for decoding this trial to trial variability in song structure, ultimately providing a predictive readout of impending behaviors.

Sequential activation of neurons is thought to provide computational advantages for encoding temporal information associated with episodic memories or behavioral sequences (*Fiete et al., 2004*; *Kumar et al., 2010*; *Rajan et al., 2016*). Neural sequences in HVC provide one of the cleanest examples linking brain activity with a naturally learned and volitionally produced skilled motor behavior (*Fee et al., 2004*). Our study provides a glimpse of how these sequences emerge through temporally coordinated transitions within a potentially hierarchically organized network and suggests a general framework for initiating the production of skilled motor behaviors.

# Materials and methods

## Key resources table

| Reagent type (species) or resource | Designation | Source or reference | Identifiers | Additional information |
|---|---|---|---|---|
| Strain, strain background (adeno-associated virus) | AAV9.CMV.HI.eGFP-Cre.WPRE.SV40 | James M. Wilson | Addgene viral prep # 105545-AAV9; http://n2t.net/addgene: 105545; RRID:Addgene_105545 | |
| Strain, strain background (adeno-associated virus) | AAV9.CAG.Flex.G CaMP6s.WPRE.SV40 | *Chen et al., 2013* | Addgene viral prep # 100842-AAV9; http://n2t.net/addgene: 100842; RRID:Addgene_100842 | |
| Strain, strain background (adeno-associated virus) | AAV9.CAG.GCaMP6s. WPRE.SV40 | *Chen et al., 2013* | Addgene viral prep # 100844-AAV9; http://n2t.net/addgene: 100844; RRID:Addgene_100844 | |
| Commercial assay or kit | Miniature Microscope | Inscopix | https://www. inscopix.com/nvista | |
| Software, algorithm | Matlab | Mathworks | http://www.mathworks. com/products/matlab/; RRID:SCR_001622 | |
| Software, algorithm | Calcium Analysis | *Peters et al., 2014* | | |
| Software, algorithm | CNMF | *Pnevmatikakis et al., 2016* | https://github.com/ epnev/ca_ source_extraction | |

## Software and data availability

All custom analysis codes and calcium imaging data are publically available as a Github repository (https://github.com/TRobertsLab/HVCRA_PreparatoryActivityData) (*TRobertsLab, 2019*; https://github.com/elifesciences-publications/HVCRA_PreparatoryActivityData).

## Animals

Experiments described in this study were conducted using adult male zebra finches and Bengalese finches (>90 days post hatch). During experiments, birds were housed individually in sound-attenuating chambers on a 12/12 h day/night schedule and were given ad libitum access to food and water. All procedures were performed in accordance with established protocols approved by Animal Care and Use Committee's at UT Southwestern Medical Centers, Texas Christian University, Emory University, and the Korea Brain Research Institute.

## Defining song and Peri-Song behavior

Zebra finch song is composed of a song motif which contains a stereotyped set of song syllables. The song motif is often produced two or more times in immediate succession – this is referred to as a song bout. Song bouts are often preceded by one or more introductory syllables. When a bird strings together more than one song bout, this is referred as a song phrase. In this study we defined song bouts separated by less than 2 s of silence as being part of a single song phrase, while any singing following silent gaps greater than two seconds from the last song phrase were considered as the beginning of a new song phrase. For analysis of pre-song activity we only analyzed data in which birds were not singing for a minimum of 5 s prior to production of their first introductory note or song motif. For analysis of post-song activity we only analyzed data from birds that did not engage in further singing for a minimum of 5 s following production of the last syllable in their song phrase or bout.

## Viral vectors

The following adeno-associated viral vectors were used in these experiments: AAV2/9.CAG.Flex. GCaMP6s.WPRE.SV40 (University of Pennsylvania Vetor Core; Addgene Catalog #: 100842-AAV9), AAV2/9.CMV.EGFP.Cre.WPRE.SV40 (University of Pennsylvania Vector Core; Addgene Catalog #: 105545-AAV9) and AAV2/9.CAG.GCaMP6s.WPRE.SV40 (University of Pennsylvania Vector Core; Addgene Catalog #: 100844-AAV9). All viral vectors were aliquoted and stored at −80°C until use.

## Imaging equipment

Head-mounted miniaturized fluorescent microscopy in freely behaving singing birds was conducted with an nVista system (Inscopix). Two-photon microscopy was conducted with a commercial microscope (Ultima IV, Bruker) running Prairie View software using a 20x (1.0 NA) objective (Zeiss) with excitation at 920 nm (Mai Tai HP DS, Newport). Two-photon and initial single photon imaging was conducted in lightly anesthetized head-fixed animals. Two-photon and single-photon images of HVC were acquired through the cranial window using a sCMOS camera (QImaging, optiMOS) and these images were used to guide placement of the baseplate for the miniaturized single-photon microscope (Inscopix). CAD files for head holders and stereotaxic devices are available upon request.

## Stereotaxic surgery

All surgical procedures were performed under aseptic conditions. Birds were anesthetized using isoflurane inhalation (~1.5–2%) and placed in a stereotaxic apparatus. Viral injections were performed using previously described procedures (*Roberts et al., 2012*, *Roberts et al., 2017*) at the following approximate stereotaxic coordinates relative to interaural zero and the brain surface (rostral, lateral, depth, in mm): HVC (0, 2.4, 0.1–0.6); and RA (−1.0, 2.4, 1.7–2.4). The centers of HVC and RA were identified with electrophysiology. For calcium imaging experiments, 1.0–1.5 µL of Cre-dependent GCaMP6s was injected at three different sites into HVC and 350 nL of Cre was injected into RA. Viruses were allowed to express for a minimum of 6 weeks before a cranial window over HVC was made.

## Cranial windowing, Imaging and Baseplate Implantation

Briefly, a unilateral square craniotomy (~3.5×3.5 mm) was created over HVC and the dura was removed. A glass coverslip was cut to match the dimensions of the craniotomy and held in place with a stereotaxic arm as Kwik Sil was applied to the edges of the cranial window. Dental acrylic was applied over the Kwik Sil and allowed to slightly overlap with the glass coverslip to ensure the window would not move and would apply the appropriate amount of pressure to the brain. An aluminum head post was affixed to the front of the bird's head to enable head-fixed imaging under the 2-photon microscope and to enable head-fixation for baseplate implantation. Following verification of labeling, identification of HVC boundaries, and high-resolution images of neurons under the 2-photon microscope, the bird was lightly anesthetized with isoflurane and the miniaturized fluorescent microscope (Inscopix) was placed over the cranial window. The field of view that matched the 2-photon images was identified and the focal plane that enabled the largest number of neurons to be in focus was selected. Dental acrylic was used to fix the baseplate in the desired position and any exposed skull was covered with dental acrylic. Once the dental acrylic dried, the microscope was removed from the baseplate and the bird was allowed to recover overnight. About 30 min before the birds' subjective daytime, the microscope was attached to the counterbalance (Instech) with enough cable to allow the bird to move freely throughout the cage. The microscope was then secured to the baseplate with a setscrew. The bird was allowed to wake up and accommodate to the weight of the microscope over the next 2–3 days. After the end of imaging experiments, the baseplate was removed and stereotactic injections of AlexFluor 594 were made into Area X. After 1 week, birds were given an overdose of Euthasol, perfused and tissue was sectioned.

## GCaMP6s imaging using a miniaturized fluorescent microscope

The miniaturized fluorescent microscope (Inscopix) was not removed following successful baseplate implantation and remained attached to the birds' head until either the cranial window closed or 7–10 days had passed. The counterbalance was adjusted based on the observed behavior of the bird and its ability to move freely. The female was not housed in the cage with the male bird, but instead was introduced to the males during a minimum of 3 morning and afternoon sessions to evoke directed song. Video recording was first started followed by 5 to 10 s of spontaneous recording with the miniaturized fluorescent microscope. The female bird was placed in the cage as quickly and as with little disruption as possible for each session. If the male bird did not sing within a minute of the females' presence, the session was stopped, and the female was removed. All trials were recorded on video, and audio was recorded using Sound Analysis Pro (SAP) software and the HD video camera microphone. Calcium imaging was performed at 30 frames per second (fps), at 1080 × 1920 resolution, Gain was set to 4, and Power was set to 90% for all birds, behavioral videos were collected at 24 fps. Calcium imaging data and behavioral data was synchronized using start of calcium imaging on a frame by frame basis.

## Calcium image processing and analysis

Calcium images were collected using the miniaturized fluorescent microscope developed by Inscopix (*Ghosh et al., 2011*). Video recordings of birds behavior and singing were manually synchronized on a frame by frame basis to the onset of calcium imaging, visualized by turning of the blue LED on the miniscopes. For processing of calcium imaging data the FOV (Field of View) was spatially cropped to exclude pixels that did not include neurons or observable changes in fluorescence. Next, the preprocessing utility within the Mosaic data analysis software was used to spatially bin the images by a factor of 2 to reduce demands on computer memory and enable faster data processing. The TurboReg implementation within Mosaic was used to perform motion correction. A reference image was created using a maximum intensity projection of the dataset and the images were aligned in the x and y dimension to the reference image. Imaging datasets with translational motion greater than 20 pixels in either the x or y dimensions were excluded from further data analysis. Post-registration black borders were spatially cropped out. The resulting spatially-cropped, preprocessed, and motion corrected calcium imaging datasets were exported for further analysis in custom Matlab scripts.

We performed ROI-based analysis on the motion-corrected calcium imaging datasets using previously described methods (*Peters et al., 2014*) (see *Source code 1–4*). ROIs were manually drawn

around identifiable soma and a secondary ROI that extended six pixels around the boundaries of the neuronal ROI was used to estimate background fluorescence (i.e. neuropil or other neurons). The pixel values were averaged within the neuronal and background ROIs, and background fluorescence signal was subtracted from neuronal signal. An iterative procedure using custom Matlab scripts were used to estimate baseline fluorescence, noise, and active portions of the traces (*Peters et al., 2014*). A subset of calcium images were re-analyzed using previously described constrained non-negative matrix factorization (CNMF) methods, but calcium fluorescence traces were identical to the traces pulled out by the ROI-based analysis (*Pnevmatikakis et al., 2016*). Calcium traces generated by ROI-based analysis were further deconvolved to produce inferred calcium traces using the pool adjacent violators algorithm (PAVA) (*Friedrich et al., 2017*). The deconvolved calcium traces were normalized to values between 1 and 0 to enable visualization of activity across different neurons during the same trial. Calcium transients that were 3 SD above baseline activity were recorded as events. The corresponding onset times and the rise times to peak fluorescence of individual calcium transients were correlated with synchronized behavior.

All calcium events were first categorized as falling into peri-song or song behavioral epochs on a frame by frame basis. Peri-song was limited to the 5 s period before the onset of vocalizations, including introductory notes, and the 5 s period after the offset of the last syllable. These event counts were used to assign a Phrase index to all imaged neurons. Neurons that had fewer than two calcium events recorded over a day of singing were excluded from further analysis because sparsity of calcium events could spuriously identify neurons as peri-song or song exclusive. We combined the number of calcium events from neurons imaged across multiple trials during the same day. Neurons imaged across multiple days were treated as unique neurons. The phrase index was calculated as a ratio of the total number of song events imaged from a neuron during a day subtracted by the number of peri-song events to the total number of calcium events. This bounded the phrase index to values of −1 (peri-song exclusive) and +1 (song exclusive). We used the phrase index to examine the timing properties of neurons active only during peri-song, song, or both behavioral epochs.

To examine the distribution of calcium events, we generated histograms with bin sizes of 200 ms. Peri-song event rates were first calculated in 100 ms bins and then a plotted using a 1 s moving average window that reaches a minimum of 500 ms at boundaries (−5,+5 s, and at song onset and offset). The average event rate and standard deviation was calculated using all pre and post phrase event rates from all birds.

## Fluorescence analysis across intervals

Average fluorescent changes were measured for each neuron across baseline, pre-song, post-song, and song behavioral periods. Baseline was defined as a behaviorally quiet period covering 5 s of fluorescent activity that was ≥10 s removed from periods of singing or calling. Pre-song was 5 s before phrase onset and post-song was 5 s after phrase offset. The background subtracted fluorescent traces were used to measure average fluorescence across the above intervals for all phrases and all birds. Averaged fluorescent values were than normalized to the average fluorescence measured during song.

## Comparison of Signal-to-Noise ratio and calcium event peak magnitudes

We measured the SNR of events occurring during song for a subset of pan-song and song neurons. The SNR was calculated as a ratio of peak fluorescence for each song event per neuron to the average fluorescence from baseline period within the trial (as above, 5 s of fluorescent activity that was ≥10 s removed from periods of singing or calling). We determined the average SNR for each neuron and examined differences between pan-song and song neurons.

Peak magnitudes during peri-song and song periods of pan-song neurons were measured using normalized deconvolved fluorescent traces. The peak values for each pan-song neuron (with phrase indices between −0.18 to +0.18) during peri-song and song periods were used to evaluate potential differences between calcium events occurring outside of song versus during song.

## Neural recordings

Multiunit recordings of HVC neurons were collected from three adult (>90 days old) male Bengalese finches. All procedures were approved by the Korea Brain Research Institute. An array of 16 tungsten microwires (175 μm spacing, OMN1005-16, Tucker Davis Technologies) was implanted into left HVC. The location of HVC was identified by searching for spontaneous spike bursts and for antidromic response to stimulation in RA. The extracellular voltage traces of all channels from birds singing alone (without presentation of female) were amplified and recorded with an interface board (RHD2132, Intan Technologies) at a sampling rate of 25 kHz. The interface board was tethered to a passive commutator (Dragonfly Inc) via a custom-made light-weight cable. In total we obtained HVC recordings from 35 electrode channels in three birds (15, 7, and 8 channels, respectively), all of which showed spontaneous bursts typical of HVC neurons. The impedance of successful electrodes was around 100–300 kΩ.

Recorded signals were bandpass filtered (0.3–5 kHz) and negative signal peaks exceeding 4 SD from spontaneous activity (spontaneous activity was measured from time points greater than 10 s from the nearest song bout) were interpreted as multi-unit spikes. In total 38, 62, and 212 song onsets, and 42, 23, and 280 offsets were identified in these birds, respectively. We produced firing rate-traces from each electrode channel with 10 ms resolution and averaged them across song renditions. After smoothing with a 500 ms moving average window, the averaged firing rates were normalized between 0 and 1 to enable comparisons of recordings from different channels and to obtain the general trend of onset- and offset-related firing across channels and birds.

The significance of activity elevation during pre-song, song, and post-song periods from the baseline was tested by Wilcoxon signed-rank test with significant level at 0.05 after Bonferroni correction for multiple comparison. Onset of pre-song and offset of post-song activity were estimated for each channel as the smoothed spike rate trajectory was exceeded a threshold which was defined as mean +2 SD of the spontaneous spike rate.

Single-unit and multiunit recordings of RA neurons were collected from six adult (>140 days old) male Bengalese finches as described previously (*Tang et al., 2014*). All procedures were approved by the Emory University Institutional Animal Care and Use Committee. Briefly, an array of four or five high-impedance microelectrodes was implanted above RA. We remotely advanced the electrodes through RA using a miniaturize motorized microdrive and recorded extracellular voltage traces as birds produced undirected song (i.e., no female bird was present). We used a previously described spike sorting algorithm to classify individual recordings as single-unit or multiunit (*Sober et al., 2008*). In total, we recorded 19 single units (multiunit recordings were not analyzed further in this study). Based on the spike waveforms and response properties of the recordings, all RA recordings were classified as putative projection neurons (*Leonardo and Fee, 2005*; *Sober et al., 2008*; *Spiro et al., 1999*).

## Analysis of chronic recording data

To analyze the variation in inter-spike-interval (ISI) in different time periods (*Figure 5*), we restricted our analysis to cases in which we collected at least one recording that included the relevant song epoch. 'Spontaneous' epochs were sampled from neural activity recorded more than 10 s after the nearest song bout. 'Pre-song' activity was sampled from between 2.5 and 0.5 s prior to the first song syllable or introductory note. 'Song' activity was sampled from the onset of the first song syllable until the offset of the last syllable in a bout. 'Post-song' activity was sampled from between 0.5 and 2.5 s after the offset of the last syllable in a bout. In some cases, we did not have sufficient data available from all epochs for all neurons (note the variation in the number of neurons included in the analyses shown in *Figure 5d*).

## Air sac recording procedures

Subsyringeal air pressure was recorded from six adult male zebra finches in directed singing conditions. Directed song was defined as a female presented in an adjacent cage during a two-hour recording period. Data from four of the birds were re-analyzed from a previously published study (*Cooper and Goller, 2006*) and data from two additional birds were collected to replicate the effects observed in the previously collected data (*Cooper and Goller, 2006*). As described in (*Secora et al., 2012*), each bird was accustomed to carrying a pressure transducer that was held in

place on the bird's back with an elastic band (*Secora et al., 2012*). To facilitate relatively free lateral and vertical movement in the cage, the weight of the transducer was offset by a counter-balance arm. Subsyringeal air pressure surgery was performed after birds sang while carrying the pressure transducer. Prior to insertion of the air pressure cannula, animals were deeply anesthetized as verified by an absence of a toe-pinch response. A small opening in the body wall below the last rib was made with a fine pair of micro-dissecting forceps, and a flexible cannula (silastic tubing, OD 1.65 mm, 6.5 cm length) was inserted into the body wall and suture was tied around the cannula and routed between the 2$^{nd}$ and 3$^{rd}$ ribs to hold it in place. The skin was sealed to the cannula with tissue adhesive (Nexaband). The free end of the cannula was attached to the pressure transducer. This allowed for measurement of relative subsyringeal air pressure changes inside the thoracic air sac before, during, and after spontaneously generated song events. Birds were monitored following surgery until they perched in the recording chamber.

The voltage output of the pressure transducer was amplified (50–100 x) and low-pass filtered (3 kHz cutoff; Brownlee, Model 440, Neurophase, Santa Clara, CA). Respiration was recorded for five seconds prior to and following singing epochs using a National Instruments analog-to-digital conversion board (NI USB 6251, Austin, TX) controlled by Avisoft Recorder software (Avisoft Bioacoustics, Berlin, Germany). Data were collected in wav file format, 16 bit resolution, with sampling rates varying from 22.05 to 40 kHz. Songs were selected for analysis that contained at least 3 s of uninterrupted quiet respiration prior to and following song. Songs that were preceded by calls, drinking, defecation, or movement-related activity were excluded from the analysis.

### Air sac data classification

Air pressure was analyzed as respiratory cycles, which was defined as an inspiration followed by expiration. The onset of inspiration was identified as subambient air pressurization and the return to ambient pressure following the expiratory phase of the cycle. The cycle duration (s), duty cycle (% time spent in the expiratory phase of respiration), and average rectified amplitude (a.u.) was calculated for each cycle. Song respiration was analyzed prior to song onset and following song termination. Song onset was defined as the inspiration preceding the first introductory note; using this marker, the onset time for each respiratory cycle in the pre-song recording period was determined. The conclusion of song was defined as the termination of the expiration generating the last song syllable in the bird's song bout. The timing of the respiratory cycles following song were identified relative to the song termination marker.

### Statistical analyses of respiratory data

For statistical analyses of the respiratory data, each bird contributed a single average value for each measured parameter (cycle duration, duty cycle, average amplitude). A repeated measures ANOVA was used to determine how respiration changes prior to and following song. For each bird, ten to twenty songs were identified for the statistical analysis (see above for criteria). The average for the pre- and post-song (3–5 s) for each measured parameter for each bird was calculated. To evaluate the time course of change in respiratory patterns preceding and following song, the average for one second bins for each bird were used in the repeated measures ANOVA. In cases where the assumption of sphericity was violated, the Greenhouse-Geisser correction for the *degrees of freedom* was used. All *p* values reported are based on this correction. An *a priori* alpha level of 0.05 was used for determining statistical significance.

## Acknowledgements

The authors thank Joseph Takahashi for generous use of a miniscope for calcium imaging experiments, Jennifer Holdway, Maaya Ikeda, Devin Merullo, Brad Pfeiffer, Alynda Wood, and Lei Xiao for comments on the manuscript and discussions, the Genie Project (Janelia HHMI) for development of calcium indicators, Takaki Komiyama and Andrew Peters for analysis codes, Andrea Guerrero for laboratory support and animal husbandry, and Jacque Dukes for assistance with data analysis.

## Additional information

### Funding

| Funder | Grant reference number | Author |
|---|---|---|
| National Institute of Neurological Disorders and Stroke | R01NS108424 | Brenton G Cooper<br>Richard HR Hahnloser<br>Todd F Roberts |
| National Institute on Deafness and Other Communication Disorders | R01DC014364 | Todd F Roberts |
| National Science Foundation | IOS-1457206 | Todd F Roberts |
| Swiss National Science Foundation | 31003A_127024 | Richard HR Hahnloser |
| Swiss National Science Foundation | 31003A_156976 | Richard HR Hahnloser |
| National Institute of Neurological Disorders and Stroke | R01NS084844 | Samuel J Sober |
| National Institute of Neurological Disorders and Stroke | R01NS099375 | Samuel J Sober |
| Korea Brain Research Institute Basic Research Program | 18-BR-01-06 | Satoshi Kojima |

The funders had no role in study design, data collection and interpretation, or the decision to submit the work for publication.

### Author contributions

Vamsi K Daliparthi, Conceptualization, Software, Formal analysis, Investigation, Writing—original draft, Writing—review and editing; Ryosuke O Tachibana, Brenton G Cooper, Formal analysis, Investigation, Writing—review and editing; Richard HR Hahnloser, Todd F Roberts, Conceptualization, Supervision, Funding acquisition, Investigation, Writing—review and editing; Satoshi Kojima, Supervision, Investigation, Writing—review and editing; Samuel J Sober, Formal analysis, Funding acquisition, Investigation, Writing—review and editing

### Author ORCIDs

Vamsi K Daliparthi https://orcid.org/0000-0001-6358-729X
Ryosuke O Tachibana https://orcid.org/0000-0002-4766-4504
Brenton G Cooper https://orcid.org/0000-0002-3572-8822
Richard HR Hahnloser http://orcid.org/0000-0002-4039-7773
Satoshi Kojima https://orcid.org/0000-0002-0753-4238
Samuel J Sober http://orcid.org/0000-0002-1140-7469
Todd F Roberts https://orcid.org/0000-0002-0967-6598

### Ethics

Animal experimentation: Experiments described in this study were conducted using adult male zebra finches and Bengalese finches ( >90 days post hatch). During experiments, birds were housed individually in sound-attenuating chambers on a 12/12 h day/night schedule and were given ad libitum access to food and water. All procedures were performed in accordance with established protocols approved by Animal Care and Use Committee's at UT Southwestern Medical Centers (2016-101562), Texas Christian University, Emory University, and the Korea Brain Research Institute. Research conducted by our colleagues in Korea was under IACUC-15-00028 and research at Emory was under 2003538.

### Decision letter and Author response

Decision letter https://doi.org/10.7554/eLife.43732.052

Author response https://doi.org/10.7554/eLife.43732.053

## Additional files

### Supplementary files

• Source code 1. Source code for calcium trace extraction.
DOI: https://doi.org/10.7554/eLife.43732.036

• Source code 2. Source code for calcium trace baseline estimation.
DOI: https://doi.org/10.7554/eLife.43732.037

• Source code 3. Source code for creating ROIs in imaging datasets.
DOI: https://doi.org/10.7554/eLife.43732.038

• Source code 4. Source code containing helper functions for trace extraction.
DOI: https://doi.org/10.7554/eLife.43732.039

• Supplementary file 1. Summary of behavioral data set for in vivo calcium imaging experiments. *Male did not sing despite having a female present. **Male was actively calling during this trial. ***Male did not sing despite being in the presence of a female, however, the bird does perform introductory notes.
DOI: https://doi.org/10.7554/eLife.43732.040

• Supplementary file 2. Table describing categories of neurons and the functional definitions used in this study.
DOI: https://doi.org/10.7554/eLife.43732.041

• Audio file 1. *Figure 2A*: Inset audio.
DOI: https://doi.org/10.7554/eLife.43732.042

• Audio file 2. *Figure 2C*: Inset audio1.
DOI: https://doi.org/10.7554/eLife.43732.043

• Audio file 3. *Figure 2C*: Inset audio 2.
DOI: https://doi.org/10.7554/eLife.43732.044

• Audio file 4. *Figure 2E*: Inset audio 1.
DOI: https://doi.org/10.7554/eLife.43732.045

• Audio file 5. *Figure 2E*: Inset audio 2.
DOI: https://doi.org/10.7554/eLife.43732.046

• Audio file 6. Directed singing 160046 audio inset 1.
DOI: https://doi.org/10.7554/eLife.43732.047

• Audio file 7. Directed singing 162048 audio inset1.
DOI: https://doi.org/10.7554/eLife.43732.048

• Audio file 8. Directed singing 162048 audio inset 2.
DOI: https://doi.org/10.7554/eLife.43732.049

• Transparent reporting form
DOI: https://doi.org/10.7554/eLife.43732.050

### Data availability

All data generated or analysed during this study are included in the manuscript and supporting files. Source data files have been included for the following main figures: 1F; 2B; 2D; 3G; 3H; 3I; 3J; 4G; 5D; 6B-E. All the data has been compiled into a single excel file, with the corresponding data represented in different sheet tabs. Matlab files used for calcium imaging analysis, specifically for selecting ROIs and filtering calcium traces, have also been included.

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
