## [Decision Letter]

Thank you for submitting your article "Transitioning between preparatory and precisely sequenced neuronal activity in production of a skilled behavior" for consideration by *eLife*. Your article has been reviewed by Eve Marder as the Senior Editor, Erich Jarvis as the Reviewing Editor, and two reviewers. The following individual involved in review of your submission has agreed to reveal his identity: Franz Goller (Reviewer #2).

The reviewers have discussed the reviews with one another and the Reviewing Editor has drafted this decision to help you prepare a revised submission.

Summary:

This study by Daliparthi et al., presents an interesting new finding, that a subpopulation of neurons in the motor forebrain nucleus HVC of songbirds has exclusive premotor and/or postmotor activity, that does not appear to correlate with the actual vocalizations produced. To make these discoveries, the authors used GCaMP calcium imaging of HVC neurons that project to RA song nucleus of singing zebra finches, multiunit electrophysiological recordings in HVC of Bengalese finches, and respiratory activity in air sacs that is consistent with a pre-motor preparation before singing. Their interpretation is that the peri-song activity in HVC (as they call it) could be responsible for initiating and recovering singing behavior, or motor planning of the behavior. This is a technical tour de force study, especially the Ca^2+^ imaging in a specific neurons type, of awake singing birds.

Although the reviewers were excited with the approach used, there were mixed opinions on the novelty of the findings. The main novel finding that the authors say that they have is pre and post singing activity in the HVC song nucleus, and their interpretation that this cannot be controlling the motor act of producing song syllables/sequencing. It seems clear that there is a mismatch between what the authors say is published in the literature, and what is in the literature. There are studies that have shown pre and post singing neural firing, not just in HVC or HVC_RA_ projecting neurons, but in other song system neurons as well. Where the real novelty may lie, is how much more have these authors learned about this? This study may be one of the first to find neurons that fire exclusively before or after singing, but not during. And if this is the case, what insights have the authors gained that others have not gained in the past studies? But this is not what the authors have done. Rather, they appear to present a misinterpretation of their findings relevant to the current literature. This results in the authors' overselling their story and underselling the past literature, taking a strawman approach. Further, such a big difference may not be necessary for the study to be considered important. It appears to the editors that the paper will require significant revisions and more balanced justifications to be accepted.

Essential revisions:

The authors must carefully reconsider and present what is novel relevant to the current literature, and their interpretation of those findings (particularly in reference to Okubo et al., 2016; Day et al., 2009; Goldberg et al., 2010 and Kao et al., 2008). The authors first say that studies in the past have not found pre-song activity in HVC, and that it was thought to only function at the moment of singing for sequences of vocalizations, citing single unit studies. Then they more correctly claim later in the paper, that electrophysiology studies have found premotor activity in HVC, using multi-unit recordings. What the authors need to do is be fair to the field of the neurobiology of motor behavior of song behavior, in particular, and mention a well-known principle in neuroscience, that neural activity in brain areas that controls movement of muscle activity precedes the movement/muscle activity. This is one reason why HVC was called a premotor nucleus.

The authors introduced their topic in a very wide context of neurons controlling sequencing behavior in the brain, across vertebrates. But then neither in the Results section or Discussion section did they interpret their findings in that wider context. So, the reader is left with a myoptic view of the HVC song nucleus, and nothing of relevance to anything else. The authors should in the discussion say how they interpret their findings in the control of learned movements generally. HVC has been proposed to function like cells in the mammalian motor cortex (layer III neurons of M1 for example Pfenning et al., 2014) or premotor cortex (Brocas area for example Bolhuis et al., 2010). Based on these comparisons, would the authors propose or predict that mammalian motor cortex show preparatory activity seconds before a movement is performed? Hasn't such analyses been undertaken already in mammalian motor cortex for movement, at the multiunit and single unit levels? If the authors are going to brush broad strokes in their Introduction, they must complete that in the Discussion section.

The authors look like they combined multiple studies from different labs without an initial plan for those studies to be combined. What this leads to are different studies with different variables and definitions. This includes different definitions of a song bout, song phrase, pre- and post-singing times for defining a bout and for analyses; different species (zebra finch vs Bengalese finch); and different social context (with a female or singing alone). It is well known that these factors make a difference in analyses and neural activity in the brain. The authors need to be more consistent in their behavioral definitions, pre- and post-singing time periods across studies, and the social context used. They should consider doing some of their analyses over again with the same timing of 1, 2, or 5 seconds pre and post singing to make them more consistent, and not one set of times (5 seconds) for the GCaMP imaging and another (2 seconds) for the multiunit electrophysiology. Also, in some of their analyses they seem to not include introductory notes in the bouts/phrases, and sometimes do. If they do not, then they could get an artificial error of thinking they found activity before singing when it is really activity occurring during singing the introductory notes. In the video the HVC neurons seem to become active during production of the introductory notes.

[Editors' note: further revisions were requested prior to acceptance, as described below.]

Thank you for resubmitting your work entitled "Transitioning between preparatory and precisely sequenced neuronal activity in production of a skilled behavior" for further consideration at *eLife*. Your revised article has been favorably evaluated by Eve Marder (Senior Editor), and two reviewers. We apologize for the time this taken, but we were waiting for input from one of the relevant parties and now are just proceeding without that input.

Both reviewers feel that the manuscript has been substantially improved but one of the reviewers still is concerned about some issues.

Two important (yet addressable) concerns remain.

1) A previous concern was how bout onsets were defined and how cage noise was defined. The authors responded that the reviewers should 'rest assured' that the bout onsets are correctly labeled.

The spectrograms in Figure 2A,C,E and many of the supplementary Figures (e.g. Birds 162048, 160046) still all have what look like vocalizations prior to the red lines marking the bout onset. Specifically, there are apparent harmonics in the spectrogram in the 2-8khz range that look a lot like calls/intro notes in both Figure 2C,E under areas called cage noise. Cage noise, like song, can be broadband, but unlike song has lots of power concentrated at <1kHz and – also unlike song, rarely has nice harmonics with fundamentals at ~kHz ranges. But the cage noise parts of most spectrograms does not have this spectral profile – unless the spectrogram is being bandpass filtered in a way that is not find mentioned in the Materials and methods section.

Also, in the supplemental figure showing Directed Singing 162048, the sound trace underneath the mic voltage signal is totally flat for the first seconds and the HVC activity is totally silent. Then there is a sudden onset of power in the mic signal, which by eye looks to be totally synchronous with the first HVC_RA_ transient – the mic signal is then noisy for the next several seconds when there is HVC activity. A similar pattern is observed in 160046. Thus, in these examples HVC 'pre-bout' activity is associated with noises on the mic signal (reported to be cage noise). Overall, HVC activity is rarely observed when the mic signal is flat and is much more commonly observed when there is time-aligned power on the mic signal (called cage noise).

While I hesitate to be to nit-picky about this, the entire paper is about pre- and post-bout Ca^2+^ transients. Yet most spectrograms appear (by eye) to have vocalizations preceding and following defined bout onsets and offsets. Since we know that HVC_RA_ neurons can discharge during calls and intro notes, a lot depends on the authors totally nailing beyond a shadow of a doubt what constitutes a bout onset and what constitutes peri-bout concentrated cage noise.

Guidance: One way to resolve this issue would be to simply plot beneath the unfiltered mic signal a mic signal that is equal to the ratio of power in the 0.2-1 Hz range to the power in the 2-8 kHz range. Cage noise is easily distinguishable from real vocalizations by its excess power at low (<1kHz) frequencies. To address this specific reviewer concern, expanded views of the spectrograms highlighted in the comments above would also help.

Whatever the authors do, the paper would be stronger – and better for posterity – if cage noise was convincingly/objectively defined and clearly distinguished from calls, and if the showcased examples of peri-bout HVC Ca^2+^ transients were not compromised by what many readers are bound to think are spectral elements in peri-bout periods that look like vocalizations.

2) While the authors have done a much better job in the introduction and discussion contextualizing their findings in the literature, in some places they are still mis-characterizing past work. I make some recommendations below that I hope will be considered.

Subsection “Peri-Song Activity in Populations of HVC_RA_ Neurons”: The authors claim that the pre-bout burst in HVC is a "newly discovered activity profile." But that's not accurate. In fact, the very first example of a singing related projection neuron in Picardo et al., (Figure 1E) is a neuron that exhibits a pre-bout transient that precedes bout onset by several seconds – just like examples in this study. If the authors want to counter that the Picardo study imaged both HVC_X_ and HVC_RA_ neurons while this one specified HVC_RA_ neurons; they may be technically accurate but it's still misleading to report a 'new activity profile' – especially to someone who is familiar with the Picardo work (which includes pre-bout activity seconds in advance of the bout), the Okubo and Rajan papers which also include pre-bout activity.

Subsection “Peri-Song Activity in Populations of HVC_RA_ Neurons”: I think including the word 'exclusively' is unnecessary and not quite accurate description of the papers cited. Just because a paper focuses entirely on the role of HVC_RA_ activity in patterning vocalizations does not mean that the paper requires HVC_RA_ activity to discharge exclusively during vocalization.

Subsection “Peri-Song Activity in Populations of HVC_RA_ Neurons”: The authors wonder why previous studies did not observe peri-bout activity. Yet other studies did observe peri-song activity; just less of it. Again, see Picardo et al., Figure 1; Okubo et al., etc, Rajan…

Subsection “Peri-Song Activity in Populations of HVC_RA_ Neurons”: The authors' explanations/conjectures here about why potentially pre- and post-bout discharge was 'missed' are too speculative and not quite accurate. The triggering with a buffer is not how ephys studies of HVC were done. In original experiments they were done with audio monitors so sparse discharges would not be missed. Antidromic identification also does not bias for discharge during singing (thus the discovery that many HVC_RA_ neurons are silent). And during intracellular recordings from HVC in awake, singing birds – every recording is so precious and hard-earned and eyeballed in real time on a computer/oscilloscope that there is no way that pre-bout discharge would have been missed. In these studies, half of the HVC_RA_ neurons were either silent or active during introductory notes (though this was not emphasized in the papers).

A real possibility that could be included here is that Ca^2+^ transients do not equal spiking and vice versa. Dopamine 'ramps' are observed in Ca^2+^ imaging of neurons and in fiber photometry but have never been observed in DA spiking (there was a long discussion of this issue at CoSyne 2018 meeting, it is covered in recent review by Josh Berke (Berke, 2018), and is relevant to ramping signals observed in the recently posted biorxiv paper from the Witten group (https://www.biorxiv.org/content/10.1101/456194v1)).

Also, inhibition can independently regulate Ca^2+^ influx and spiking – resulting in non-congruence between spikes and GCAMP Ca^2+^ signals (nicely shown in Owen, Berke and Kreitzer, 2018);

Discussion section: Get rid of 'exclusively'.

Discussion section: Get rid of 'only'.

Discussion section: Again the statement that pre-bout Activity in HVC was not previously observed is not accurate; see Picardo et al., Figure 1.

In the interest of expediency, I have thought about a faster way that the authors could address the issue of cage noise and bout onset definition. It seems to me that if the authors simply provide the wav files that correspond to all of the spectrograms in Figure 2 and Figure 2—figure supplement 1 than any reviewer and reader could just download the song and listen to it him/herself.

[Editors' note: further revisions were requested prior to acceptance, as described below.]

Thank you for resubmitting your work entitled "Transitioning between preparatory and precisely sequenced neuronal activity in production of a skilled behavior" for further consideration at *eLife*. Your revised article has been favorably evaluated by Eve Marder (Senior Editor) and Erich Jarvis (Reviewing Editor). The manuscript has been improved but there are some remaining issues noted by the Reviewing Editor that need to be addressed before acceptance, as outlined below:

There is a remaining issue with interpretation and semantics of what is pre- and post-vocalization activity measured in HVC and the song system relative to prior studies, which is what has been causing confusion to the reviewers. From a semantic understanding, pre-song activity in HVC-RA or any neurons of the song system could be considered 1 ms, 10ms, 100ms, 1s, 10s, or 10 min before vocalizing. In all cases, activity increases occur before vocalizing. All are premotor changes in neural firing. The difference that the authors find is that some neurons continue to fire during the motif or syllable production that has been found in previous studies and others do not, the new finding of this study. These former are what the authors appear to characterize as 'during song neurons', even if they fire 1-500 ms before or after the sound itself. The later fire 500ms to 3,000ms before sound is produced, and some of these shut off during the moments of syllable production. The authors assume that the idea of the 1-500ms range of premotor activity is "baked in", and therefore they call them 'during song production/singing neurons'. But what is baked-in the neuroscience field is that there is premotor activity in motor pathways in the manuscript before sound production or other behavior production. Can the authors see the confusion here?

To be less confusing to readers, the authors should include a table that defines the different neuron categories according to their temporal firing patterns, and then use those definitions consistently throughout the paper. Example terms that need such definitions in the table are: peri-song neurons (presumably within 5-0.5s firing before or after the first or last song syllable, respectively); pre-song neurons (presumably firing ~0.5s to 1ms before each syllable or the first song syllable); post-song; pan-song, etc. The real novelty of this study is finding neurons of the peri-song neurons as defined above, and not even just HVC-RA projecting neurons, but the fact that any such neurons exist at all in the song system. The paper needs to highlight this novelty more clearly. To do so, this difference in timing of neuron population firing and pattern should be stated in the Abstract, end of Introduction and in sentences of the Results section when this topic is discussed.

More specifically, what appears most novel about the current study is that the authors find subpopulations of HVC-RA projection neurons that: (a) fire 1-3 seconds before singing starts [or after it ends]; (b) some of which turn off a few ms before or right at the time singing starts; and (c) are not correlated with as of yet identified pattern of singing. The other types of neural firing patterns, including pre- and post-song production, have been found in prior studies, and are confirmed in this study. This needs to be stated much more clearly in the manuscript.

Introduction: Making too strong of a statement that HVC-RA neurons are exclusively active during vocalization in awake birds. Even in awake birds, physiology and immediate gene studies have shown some activity to hearing songs in other species (even if birds own song) or eating (even if difficult to interpret gene expression studies). Instead of saying exclusively, how about predominantly.

Results section: The authors claim that the pre-bout burst in HVC is a "newly discovered activity profile." But that's not accurate when considering the 500 ms window of pre-bout activity. Here is where the authors would benefit by using more well-defined terminology that they show in a table as well as define the timing of what they mean by pre-bout activity.

Discussion section: It appears the Bengalese finch electrophysiology peri-song activity is within a shorter time window of several 100ms rather than 1-3 seconds as found with Ca^2+^ imaging in zebra finches. Is this a species difference, a physiology versus GCAMP sensing difference, or some other difference?

The song system diagram of Figure 1A is appears to be simplified too much. It does not show MAN, the loop with AreaX and DLM, and the feedback to RA (as well as to HVC). Such additions to the figure would not cause more confusion, and in fact would add clarity and has been what is commonly shown in birdsong diagrams for several decades now. If it would make it easier for the authors, they could use and modify one of the figures showing these connections in the adobe illustrator versions on the Jarvis Lab website: http://jarvislab.net/summary-figure-originals/ These are more polished than the hand drawn one of this paper.

As the authors wrote in one of their responses, they should mention that the variability of peri-song neuron activity on different bouts is similar to variable pre-movement activity found primate M1 neurons. Such a comparison broadens the impact of the paper.

---

## [Author Response]

The reviewers have discussed the reviews with one another and the Reviewing Editor has drafted this decision to help you prepare a revised submission.

We thank the reviewers for their comments. We have endeavored to address all of them and provide written responses to each below. We have been asked to address three essential editorial revisions to our manuscript that involve adding additional discussion of the literature to the Introduction, the Discussion section, and to provide a more seamless presentation of results for some of our data. We have directly addressed these requests outlined by the editor. We have (1) added a discussion of previous literature showing premotor activity in HVC-X, HVC-interneurons, and other portions of the song system into the Introduction section of the manuscript, (2) broadened our Discussion section in order to interpret our findings in the context of research in the mammalian motor cortex and the initiation of neuronal sequences more generally, and (3) endeavored to be more consistent in the descriptions, analysis and presentation of the research conducted across the multiple laboratories that have contributed to this study. In addition to these revisions we have carefully gone through each comment by the reviewers and editor and provide detailed replies to each.

Summary:This study by Daliparthi et al., presents an interesting new finding, that a subpopulation of neurons in the motor forebrain nucleus HVC of songbirds has exclusive premotor and/or postmotor activity, that does not appear to correlate with the actual vocalizations produced. To make these discoveries, the authors used GCaMP calcium imaging of HVC neurons that project to RA song nucleus of singing zebra finches, multiunit electrophysiological recordings in HVC of Bengalese finches, and respiratory activity in air sacs that is consistent with a pre motor preparation before singing. Their interpretation is that the peri-song activity in HVC (as they call it) could be responsible for initiating and recovering singing behavior, or motor planning of the behavior. This is a technical tour de force study, especially the Ca^2+^ imaging in a specific neurons type, of awake singing birds.

We thank the editor and reviewers for the praise of the calcium imaging approaches used here and for our findings more generally. These technically challenging experiments allowed us – for the first time – to examine population level HVC-RA neuron activity in freely-behaving zebra finches and we believe that these findings provide a new view for how precise neuronal sequences can arise in the brain.

Although the reviewers were excited with the approach used, there were mixed opinions on the novelty of the findings. The main novel finding that the authors say that they have is pre and post singing activity in the HVC song nucleus, and their interpretation that this cannot be controlling the motor act of producing song syllables/sequencing. It seems clear that there is a mismatch between what the authors say is published in the literature, and what is in the literature. There are studies that have shown pre and post singing neural firing, not just in HVC or HVC_RA_ projecting neurons, but in other song system neurons as well. Where the real novelty may lie, is how much more have these authors learned about this? This study may be one of the first to find neurons that fire exclusively before or after singing, but not during. And if this is the case, what insights have the authors gained that others have not gained in the past studies? But this is not what the authors have done. Rather, they appear to present a misinterpretation of their findings relevant to the current literature. This results in the authors' overselling their story and underselling the past literature, taking a strawman approach. Further, such a big difference may not be necessary for the study to be considered important. It appears to the editors that the paper will require significant revisions and more balanced justifications to be accepted.

In our original version of the manuscript we did not mean to imply that pre and post singing activity in other classes of neurons in HVC or other areas of the song system does not occur. We are aware of this literature and both cited and discussed it in the Results section and Discussion section of the original manuscript. We believe the confusion has arisen from the distinction between previous recordings of different classes of HVC projection neurons and interneurons, whereas our work has focused exclusively on HVC-RA neural activity. We have added a discussion of this previous literature in the Introduction and more discussion of it in the Discussion section. In addition, in the revised manuscript we have tried to clarify the writing to focus the reader on the fact that the calcium imaging is focused on HVC-RA neurons. Our original argument is that we have a novel observation of activity related to motor planning and preparation in this population of neurons.

Reviewers seem to claim that pre- and post-singing activity has previously been reported in HVC-RA neurons. We have been unable to find strong support for this in the literature. It is exceedingly difficult to record from HVC-RA neurons – this is why most people do not do this and why there are only a handful of papers that have successfully recorded from identified HVC-RA neurons. The reason they are difficult to record from is that (according to the published literature) these neurons do not fire action potentials unless the bird is singing. HVC-RA neurons are not spontaneously active, necessitating antidromic identification to identify the neurons prior to recording electrophysiological activity of these cells in vivo. Moreover, it is well documented that different classes of HVC projection neurons or interneurons have different firing patterns during singing (see for example Kozhevnikov and Fee, 2007) – therefore recordings from other classes of neurons (e.g., HVC-X, or HVC-Av) or multi-unit recordings, which are dominated by interneuron activity, cannot be used to infer the activity profile of HVC-RA neurons.

To thoroughly evaluate what is known about HVC-RA neuron activity in zebra finches we have systematically reviewed the previously published literature:

In Hahnloser, Kozhevnikov and Fee, 2002 the first recording of HVC-RA neurons in singing birds; Kozhevnikov and Fee, 2007; and in Long, Jin and Fee, 2010 and Hamaguchi, Tanaka and Mooney, (2016) the authors use intracellular recordings in sing zebra finches.

In Okubo et al., (2015) and Lynch et al., (2016) the Fee lab reports on extracellular recordings from RA-projecting and X-projecting neurons in juvenile and adult HVC. In many instances in these manuscripts they do not directly state if the neurons are identified other than to say that they think they are projection neurons. An important conclusion from these papers is that activity patterns in HVC changes drastically over development as birds learn to sing, transitioning from more rhythmic activity to less rhythmic song locked activity. This implies that activity in juvenile birds cannot be used as a measure for how the precise neural sequences exhibited in adult birds are initiated. Nonetheless, pertinent to the editors’ statement that people have shown that HVC-RA neurons are active before and after song, in the Okubo paper they describe pre and post song activity in juvenile birds.

Please note that the authors refer to these neurons as projection neurons, not HVC-RA neurons. In the parlance of this study, this means that they are grouping data from some antidromically identified neurons projecting to either Area X or to RA and some neurons that are simply presumed to be projection neurons based on their firing patterns. It is not reasonable to draw from this statement that HVC-RA projecting neurons are active immediately before and/or after song. Therefore, we have carefully gone through the data in this manuscript as well as the published dissertation of Okubo in order to better examine the idea that HVC-RA neurons have been shown to be active before and after song. In these documents there is the report of a single identified HVC-RA neuron in a 65dph juvenile bird that is active only at the offset of song bouts and it appears to be within 100-200ms of bout offset (the Okubo paper only examines activity within 300ms of the beginning or ending of song – a very different timescale than we have examined which includes the observation that activity significantly increases 2.5 seconds prior to song onset, see Figure 1E and ED Figure 2H,K,L). We did not find any mention of HVC-RA neurons being active prior to song bouts. Moreover, the data regarding bout onset and offset neurons appears to be exclusive for a juvenile bird and is not reported in adults. It should also be noted that a main point of the Okubo study is that the rhythmic firing observed in juveniles can be used to slowly build the precise neural sequences seen in adult birds. In particular, Okubo hypothesizes in both the Nature paper and the Ph.D. thesis that the bout related activity could be useful for incorporating new song syllables from the edges of song during song learning. This activity is not hypothesized to function in motor planning or motor preparation, which is conceptually different than the main point of our study.

The most comprehensive study to date documenting pre-song activity in HVC is the recent paper from Rajan, (2018). Rajan details the activity of interneurons and X-projecting neurons in HVC prior to song onset (~217-775 ms before song onset). Rajan, (2018) does not report on the activity of HVC-RA neurons; however, in the Discussion section a review of the literature is provided that appears to affirm the arguments we make here. In the Discussion section Rajan, (2018) states:

“In the current study and in previous studies, pre-bout activity of HVC_RA_ projecting neurons has not been characterized. However, it has been reported that a fraction of HVC_RA_ projecting neurons do not have any motif related activity (Hahnloser et al., 2002; Kozhevnikov and Fee, 2007; Long et al., 2010). Whether such neurons have pre-bout activity needs to be determined by recording from HVC_RA_ projecting neurons before bout onset. Given the high connection probability between HVC_RA_ projecting neurons and interneurons (Kosche et al., 2015), it is possible that a small number of HVC_RA_ neurons with increased pre-bout activity can drive increases in a large fraction of interneurons as seen in this study. Additionally, slice recordings have shown that HVC_X_ projecting neurons are also a source of excitatory input to interneurons, albeit less frequently than HVC_RA_ projecting neurons (Mooney and Prather, 2005). They could also be a source of the increased pre-bout activity seen in interneurons as the time-scales of increases in activity were similar for HVC_X_ neurons and interneurons (Figure 4). … Assuming the presence of specialized HVC_RA_ neurons with similar preparatory roles, such neurons would be very different from the HVC_RA_ neurons that are thought to encode “time” in song (Fee et al., 2004; Long et al., 2010). In addition to differences in firing patterns (“ramping-up” like activity vs. stereotyped sparse bursting), the connectivity of these neurons would also be different reflecting their different functional roles.”

Last, we also examined computational models of song production as it relates to HVC. From these models we were not able to find mention of HVC-RA neurons functioning in a capacity associated with motor preparation or motor planning. We then examined how these papers modeled the start of song/neural sequences with the idea that they could provide more general perspective on how neural sequences in HVC are hypothesized to be initiated by the field. For example, do they arise from preparatory activity among HVC-RA neurons or do they tend to be associated with moments in song and initialized by extrinsic excitatory drive (for example). From this analysis it appears that computational models also restrict themselves to the activity associated with triggering precise neural sequences at the start of song, rather than HVC-RA transitioning from phases associated with heterogenous activity for planning and preparation, spanning several seconds, prior to launching into precise song sequences. The summary of these computational models is presented in Table 1.

Table 1

ReferenceCell TypeComputational ModelSequence InitiationFiete et al., 2010HVC-RA neurons and a global inhibitory unit representing the pool of HVC interneuronsHeterosynaptic competition combined with spike-time-dependent plasticityRandom barrage of external inputLi and Greenside, 2006Either Leaky integrate-and-fire (LIF) neurons or single-compartment conductance based (HH, for Hodgkin-Huxley like) neurons.1-dimensional, homogenous, excitatory chain of nonbursting neuronsA brief high-frequency burst.Gibb, Gentner, and Abarbanel 2009HVC-RA and HVC interneurons using single compartment Hodgkin-Huxley neurons.Chain of bistable clustersBurst was initiated with a 3-4ms DC current pulse into 50% of HVC-RA neurons.Jin and Ramazanoğlu, 2007HVC-RA neuronsTwo-compartment model of HVC-RA neurons, minimal conductance-based modelBurst propagation was initiated by stimulating eight spikes in neurons in the first group of the chain, via current injection lasting 10ms.

The robust level of HVC-RA activity in the time periods leading up to song vocalization that we describe suggest that these neurons may a play a significant role in motor planning and preparation – an important distinction from the current view of HVC-RA neuron function. The reviewers are correct in saying that neural activity that might be associated with motor planning and preparation has been described in other HVC neurons and in other regions of the songbird brain. However, the preparatory activity described in our study has important distinctions from these previous studies. First, we are examining neural activity in populations of HVC-RA neurons. These neurons are necessary for production of learned song and are widely thought to carry a timing signature for song by controlling learned, sequential motor patterns. The anterior forebrain pathway is not necessary for an adult bird to produce its learned song (lesions or inactivation of lMAN, the output of the AFP, do not disrupt the birds ability to sing – this has been shown by Brainard and Doupe and the Nordeen’s among others) and lesions of HVC-X neurons (Scharff et al., 2000) suggest that HVC-X neurons (unlike HVC-RA) are not necessary for producing song. It is not known whether HVC-RA neurons are involved in motor planning or motor preparation (see text above and quoted Discussion section of Rajan, 2018). Second, we show that subpopulations of HVC-RA neurons are active on the order of seconds before song vocalization. Previous work has identified preparatory activity in other cell types and regions on much shorter timescales: HVC-X and interneurons (~217-775 ms before song onset; Rajan, 2018), subpopulations of Area-X neurons projecting to DLM (~0.56 s before song onset; Goldberg et al., 2010), LMAN neurons during undirected song (~600 to 100 ms before song onset; Kao et al., 2008), multi-unit HVC activity in juveniles with immature song (no clear time scale, Day et al., 2009), and putative projection neurons in HVC of juveniles with immature songs (~0 to 300 ms before and after song onset, Okubo et al., 2015). These previous findings are summarized in Table 2.

Table 2

ReferenceCell TypeDescription of ActivityTimingOkubo et al. 2015Putative HVC projection neurons: N= 285; 137/285 bout onset neurons; 98/285 bout offset neurons; 50/285 active both before and after bouts. 1 identified HVC-RA neuron firing at the end of bouts. At least 3 identified HVC-X neurons (2 bout onset, 1 bout offset).Significant pre and post-bout activity in the 300ms window before and after bouts. Both pre and post-bout peaks occurred between 0 to 100 ms before or after bout onset or offset, respectively.Activity hundreds of milliseconds before or after song vocalizations in mixed populations of neurons.Kao et al., 2008LMAN neurons: N= 18; elevated activity before undirected song onset. N=16; decreased activity following cessation of undirected song.In 18 neurons, “firing rate in a 500ms interval from 600 to 100ms before undirected song was greater than the average spontaneous firing rate”. In 16 neurons, firing rate “was lower than the average level of spontaneous activity” following song offset.Activity in LMAN neurons hundreds of milliseconds before or after song vocalization, but only in undirected social context.Goldberg et al., 2010Area X-DLM neurons (HF-2 neurons): N=37 neuronsIn HF-2 neurons, “firing rate gradually increased before singing (0.56 ± 0.22s), and slowly returned to baseline following singing (1.64 ± 1.2s).”Activity in AreaX-DLM neurons exhibited pre-song activity hundreds of milliseconds before song onset, and seconds after song offset.Day et al., 2009Multiunit activity in HVCElevated activity in juvenile birds before and after song vocalization.UnclearRajan and Doupe, 2013HVC-X neurons: N= 30; 12 antidromically identified, 18 putative. Putative Interneurons: N= 16Activity of putative interneurons and HVC-X neurons shown to be correlated with onset times of introductory notes.Activity time-locked to introductory note vocalization.Vyssotski et al. 2016NIF neurons projecting to HVC16 Nif-HVC neurons fire on average above baseline levels 70 ms prior to song onset and it decays to baseline about 30 ms prior to song offset.Activity is time locked to introductory note type. Authors did not look beyond 500 or 1000 ms before song onset.Rajan 2018HVC-X neurons: N=39; 13 antidromically identified, 26 putative. Putative interneurons: N=174 HVC-X neurons increased their activity (-928 ms to -129ms prior to song bout onset). 3 HVC-X neurons decreased activity -528ms to -47ms). 13 interneurons increased activity (~-674ms to -117ms).Activity before any vocalization including introductory notes.

In our study, we report pre-song neural activity in HVC-RA neurons that is 3 SD’s above baseline 2.5s prior to song onset. Indeed, a core point of our manuscript is that neurons known to sit atop the descending motor pathway necessary for birds to produce their song (HVC-RA neurons) also exhibit activity seconds prior to song onset that is consistent with a role in both motor planning and preparation and may provide a sensitive read-out of the decision to start singing. In addition, we identify for the first time HVC-RA neurons that are active before singing, but inactive during singing. This new functional population of neurons appear to play an exclusive role in motor planning and preparation. We think this is important for several reasons. First, it suggests strong commonalities with the mammalian motor system which exhibits activity associated with motor planning and preparation. Neurons in the mammalian cortex have also been found to be involved in preparation independent of performance (Economo et al., 2018), as described here in populations of HVC-RA neurons. Second, our findings indicate that we can observe the activity of a small population of neurons and potentially track the decision-making process in the seconds before a zebra finch volitionally begins to engage in courtship singing. Third, our data suggests the intriguing possibility that we can predict if the bird will sing 2-3 seconds prior to singing, a time-frame several times longer than it takes for descending motor commands to influence muscle groups associated with the motor control of song. In sum, our findings expand our knowledge of HVC-RA neuronal activity, how it relates to mammalian motor systems, and point to this network as a core network for examining how birds decide to sing. This opens an important new avenue for future studies in the field.

Essential revisions:The authors must carefully reconsider and present what is novel relevant to the current literature, and their interpretation of those findings (particularly in reference to Okubo et al., 2016; Day et al., 2009; Goldberg et al., 2010 and Kao et al., 2008). The authors first say that studies in the past have not found pre-song activity in HVC, and that it was thought to only function at the moment of singing for sequences of vocalizations, citing single unit studies. Then they more correctly claim later in the paper, that electrophysiology studies have found premotor activity in HVC, using multi-unit recordings. What the authors need to do is be fair to the field of the neurobiology of motor behavior of song behavior, in particular, and mention a well-known principle in neuroscience, that neural activity in brain areas that controls movement of muscle activity precedes the movement/muscle activity. This is one reason why HVC was called a premotor nucleus.

We have edited the Introduction of the manuscript to include a discussion of this previous work, as requested. We hope that this will better place the functional significance of our results in the broader context of the published literature.

We have added the following paragraph to the Introduction:

“Neuronal activity related to motor planning and preparation has been associated with accurate production of volitional motor movements(Churchland et al., 2010a; Svoboda and Li, 2017) but is still poorly described in the context of initiating precise neural sequences for motor behaviors, like those exhibited in HVC_RA_ neurons. Although it is not known if HVC_RA_ neurons exhibt activity related to motor planning and preparation, previous studies have identified anticipatory or preparatory activity in other classes of HVC neurons and in other regions of the songbird brain(Goldberg et al., 2010; Goldberg and Fee, 2012; Kao et al., 2008; Keller and Hahnloser, 2009; Rajan, 2018; Roberts et al., 2017). HVC contains interneurons and at least three classes of projection neurons, including neurons projecting to the striatopallidal region Area X (HVC_X_), neurons projecting to a portion of the auditory cortex termed Avalanche (HVC_Av_), and the aforementioned HVC_RA_ neurons that encode precise premotor sequences necessary for song production(Akutagawa and Konishi, 2010; Mooney and Prather, 2005; Roberts et al., 2017). Multi-unit recordings from HVC, which are typically dominated by the activity of interneurons, show increases in activity tens to hundreds of milliseconds prior to singing(Crandall et al., 2007; Day et al., 2009; Rajan, 2018). Calcium imaging from HVC_Av_ neurons and electrophysiological recordings from HVC_X_ neurons indicate that they also become active immediately prior to song onset (Rajan, 2018; Roberts et al., 2017). These data are consistent with recordings from the downstream targets of HVC_Av_ and HVC_X_ neurons. Portions of the auditory cortex (Keller and Hahnloser, 2009) and the basal ganglia pathway involved in song learning show changes in activity immediately prior to singing (Goldberg et al., 2010; Goldberg and Fee, 2012; Kao et al., 2008). Given this background, and that ~50% of HVC_RA_ neurons may not exhibit any activity during singing (Hamaguchi et al., 2016; Long et al., 2010), we sought to examine if the precise neural sequences associated with song arise as part of larger changes in activity among populations of HVC_RA_ neurons.”

We apologize for any confusion about the claims in the Introduction of our manuscript and have worked to make it as clear as possible. In our original submission we state that past studies have not found pre-song activity in HVC-RA neurons. In the Introduction of our original submission we state:

“HVC_RA_ neurons are thought to be exclusively active during vocal production, yet ~50% of recorded HVC_RA_ neurons do not exhibit any activity during singing (Hahnloser et al., 2002; Hamaguchi et al., 2016; Kozhevnikov and Fee, 2007; Long et al., 2010; Lynch et al., 2016), leaving the function of much of this circuit unresolved.”

To avoid any confusion, we have now amended this statement to read:

“HVC_RA_ neurons are thought to be exclusively active during vocal production in waking adult birds, yet ~50% of recorded HVC_RA_ neurons do not exhibit any activity during singing (Hahnloser et al., 2002; Hamaguchi et al., 2016; Kozhevnikov and Fee, 2007; Long et al., 2010; Lynch et al., 2016), leaving the function of much of the HVC_RA_ circuitry unresolved.”

HVC-RA neurons are at the top of the descending motor pathway necessary for birds to produce their song. From our reading of the literature, it was not previously known if these neurons exhibit activity associated with motor planning or motor preparation. We only cite single unit studies because those are the only studies that use antidromic identification to record from identified HVC-RA neurons and as we mentioned above, it is inappropriate to try and infer the activity of HVC-RA neurons using multi-unit recordings (biased toward interneuron activity) or recordings from other classes of HVC neurons (which have different activity profiles during song) or neurons in other regions of the song system because they are not necessary for adult song production. We hope that the additional text in the Introduction and Discussion section of our revised manuscript help to clarify distinctions between our work and what has been previously published and the relevance of this research to the field.

The authors introduced their topic in a very wide context of neurons controlling sequencing behavior in the brain, across vertebrates. But then neither in the Results section or Discussion section did they interpret their findings in that wider context. So, the reader is left with a myoptic view of the HVC song nucleus, and nothing of relevance to anything else. The authors should in the discussion say how they interpret their findings in the control of learned movements generally. HVC has been proposed to function like cells in the mammalian motor cortex (layer III neurons of M1 for example Pfenning et al., 2014) or premotor cortex (Brocas area for example Bolhuis et al., 2010). Based on these comparisons, would the authors propose or predict that mammalian motor cortex show preparatory activity seconds before a movement is performed? Hasn't such analyses been undertaken already in mammalian motor cortex for movement, at the multiunit and single unit levels? If the authors are going to brush broad strokes in their introduction, they must complete that in the Discussion section.

The Discussion section has been expanded and our research is framed in the larger context of sequencing behavior and results are interpreted in the larger context of learned motor movements.

The authors look like they combined multiple studies from different labs without an initial plan for those studies to be combined. What this leads to are different studies with different variables and definitions. This includes different definitions of a song bout, song phrase, pre- and post-singing times for defining a bout and for analyses; different species (zebra finch vs Bengalese finch); and different social context (with a female or singing alone). It is well known that these factors make a difference in analyses and neural activity in the brain. The authors need to be more consistent in their behavioral definitions, pre- and post-singing time periods across studies, and the social context used. They should consider doing some of their analyses over again with the same timing of 1, 2, or 5 seconds pre and post singing to make them more consistent, and not one set of times (5 seconds) for the GCaMP imaging and another (2 seconds) for the multiunit electrophysiology. Also, in some of their analyses they seem to not include introductory notes in the bouts/phrases, and sometimes do. If they do not, then they could get an artificial error of thinking they found activity before singing when it is really activity occurring during singing the introductory notes. In the video the HVC neurons seem to become active during production of the introductory notes.

We apologize for the lack of clarity in the original writing of the manuscript. Throughout our manuscript, in both zebra finches and Bengalese finches, we include introductory notes and calls as part of song vocalization, as our hypothesis is that HVC-RA neurons play a role in motor preparation and planning, we wanted to examine activity occurring prior to when the birds produce any vocalizations. We have verified that we include consistent definitions of song bout, song phrase, and pre- and post-singing times across the diverse array of experiments. For consistency, we report the same time window before and after song in whenever possible, and we have added text in the Results section clarifying how these time windows were chosen for different experiments. We have also included supplementary figures when available that describe multi-unit activity in HVC and single-unit activity in RA.

[Editors' note: further revisions were requested prior to acceptance, as described below.]

Both reviewers feel that the manuscript has been substantially improved but one of the reviewers still is concerned about some issues.Two important (yet addressable) concerns remain.1) A previous concern was how bout onsets were defined and how cage noise was defined. The authors responded that the reviewers should 'rest assured' that the bout onsets are correctly labeled.The spectrograms in Figure 2A,C,E and many of the supplementary Figures (e.g. Birds 162048, 160046) still all have what look like vocalizations prior to the red lines marking the bout onset. Specifically, there are apparent harmonics in the spectrogram in the 2-8khz range that look a lot like calls/intro notes in both Figure 2C,E under areas called cage noise. Cage noise, like song, can be broadband, but unlike song has lots of power concentrated at <1kHz and – also unlike song, rarely has nice harmonics with fundamentals at ~kHz ranges. But the cage noise parts of most spectrograms does not have this spectral profile – unless the spectrogram is being bandpass filtered in a way that is not find mentioned in the Materials and methods section.Also, in the supplemental figure showing Directed Singing 162048, the sound trace underneath the mic voltage signal is totally flat for the first seconds and the HVC activity is totally silent. Then there is a sudden onset of power in the mic signal, which by eye looks to be totally synchronous with the first HVC_RA_ transient – the mic signal is then noisy for the next several seconds when there is HVC activity. A similar pattern is observed in 160046. Thus, in these examples HVC 'pre-bout' activity is associated with noises on the mic signal (reported to be cage noise). Overall, HVC activity is rarely observed when the mic signal is flat and is much more commonly observed when there is time-aligned power on the mic signal (called cage noise).While I hesitate to be to nit-picky about this, the entire paper is about pre- and post-bout Ca^2+^ transients. Yet most spectrograms appear (by eye) to have vocalizations preceding and following defined bout onsets and offsets. Since we know that HVC_RA_ neurons can discharge during calls and intro notes, a lot depends on the authors totally nailing beyond a shadow of a doubt what constitutes a bout onset and what constitutes peri-bout concentrated cage noise.Guidance: One way to resolve this issue would be to simply plot beneath the unfiltered mic signal a mic signal that is equal to the ratio of power in the 0.2-1 Hz range to the power in the 2-8 kHz range. Cage noise is easily distinguishable from real vocalizations by its excess power at low (<1kHz) frequencies. To address this specific reviewer concern, expanded views of the spectrograms highlighted in the comments above would also help.Whatever the authors do, the paper would be stronger – and better for posterity – if cage noise was convincingly/objectively defined and clearly distinguished from calls, and if the showcased examples of peri-bout HVC Ca^2+^ transients were not compromised by what many readers are bound to think are spectral elements in peri-bout periods that look like vocalizations.In the interest of expediency, I have thought about an faster way that the authors could address the issue of cage noise and bout onset definition. It seems to me that if the authors simply provide the wav files that correspond to all of the spectrograms in Figure 2 and Figure 2—figure supplement 2 than any reviewer and reader could just download the song and listen to it him/herself.

We have gone through the figures referred to above and have provided high-resolution spectrograms and audio files for expanded sections before and between vocalizations. This data covers the examples shown in Figure 2 (new data is illustrated in Figure 2figure supplement 7 and associate audio files) and for files associated with directed singing trails 162048 and 160046 (expanded regions now shown in Figure 2—figure supplement 8 and associated audio files). The reviewer is correct that there appear to be harmonic structures within some of the periods before the songs illustrated in Figure 2. Based on careful monitoring of videos from the cage during the song trials from Figure 2 and based on the duration of these harmonics and lack of frequency modulation, we have identified them as female “distance calls” and “tet” calls. Distance calls and tet calls are sexually dimorphic in zebra finches. Both call types function as contact calls used during affiliative interactions, like the social interactions used in our study to elicit courtship song (ie directed song). Please see the recent paper by Julie Elie and Frederidic Theunissen (Elie and Theunissen, 2016) for a thorough description of call types in male and female zebra finches. We apologize for this oversight and have edited the relevant figures to depict where the female calls occurred.

In addition, we examined the frequency distribution of our microphone recordings and bandpass filtered them as described above by the reviewer. First, most of the power seems to be concentrated to less than 5 kHz, however, as the reviewer notes above, cage noise can be broadband and might explain how some cage noise can appear to look like harmonic structures in low-resolution spectrograms. We hope the included audio files enable readers to verify that noisy pre-song components are not vocalizations unless otherwise noted. When there are vocalizations in the audio files, they are easily discernable by ear.

2) While the authors have done a much better job in the introduction and discussion contextualizing their findings in the literature, in some places they are still mis-characterizing past work. I make some recommendations below that I hope will be considered.Subsection “Peri-Song Activity in Populations of HVC_RA_ Neurons”: The authors claim that the pre-bout burst in HVC is a "newly discovered activity profile." But that's not accurate. In fact, the very first example of a singing related projection neuron in Picardo et al. (Figure 1E) is a neuron that exhibits a pre-bout transient that precedes bout onset by several seconds – just like examples in this study. If the authors want to counter that the Picardo study imaged both HVC_X_ and HVC_RA_ neurons while this one specified HVC_RA_ neurons; they may be technically accurate but it's still misleading to report a 'new activity profile' – especially to someone who is familiar with the Picardo work (which includes pre-bout activity seconds in advance of the bout), the Okubo and Rajan papers which also include pre-bout activity.

We have changed the first part of this sentence to eliminate the phrase “these newly discovered activity profiles”. The first part of this sentence now reads, “To better characterize the activity profiles of HVC_RA_ neurons, …”.

We do however disagree with the reviewer’s interpretation of the Picardo study. In our opinion it is not possible to know if the activity profile in Figure 1E of the Picardo study shows a similar profile to what we describe in our study. First, it looks like the bird is vocalizing prior to the onset of the song. The bird appears to be producing 2 calls followed by introductory notes before the first motif and the activity of neuron #1 may well be active due to those vocalizations. Second, there is no description in the results or Discussion section of this putative pre-song activity. Therefore, it is not possible to draw conclusions about pre-song activity from this figure panel. Third, it is not known what cell type was being recorded. The study tries to focus on putative projection neurons, but this is largely determined by the activity patterns during singing. Without further support, we believe it is not appropriate to claim that panel 1E in the Picardo study provides the first description of an HVC projection neuron exhibiting pre-song activity.

Subsection “Peri-Song Activity in Populations of HVC_RA_ Neurons”: I think including the word 'exclusively' is unnecessary and not quite accurate description of the papers cited. Just because a paper focuses entirely on the role of HVC_RA_ activity in patterning vocalizations does not mean that the paper requires HVC_RA_ activity to discharge exclusively during vocalization.

We have deleted this word.

Subsection “Peri-Song Activity in Populations of HVC_RA_ Neurons”: The authors wonder why previous studies did not observe peri-bout activity. Yet other studies did observe peri-song activity; just less of it. Again, see Picardo et al., Figure 1; Okubo et al., etc, Rajan…

We have changed this sentence, so it specifies HVC_RA_ neurons and studies of adult birds. It now reads. “These results also raise questions as to why previous studies in adult zebra finches have not identified peri-song activity in HVC_RA_ neurons.”

Subsection “Peri-Song Activity in Populations of HVC_RA_ Neurons”: The authors' explanations/conjectures here about why potentially pre- and post-bout discharge was 'missed' are too speculative and not quite accurate. The triggering with a buffer is not how ephys studies of HVC were done. In original experiments they were done with audio monitors so sparse discharges would not be missed. Antidromic identification also does not bias for discharge during singing (thus the discovery that many HVC_RA_ neurons are silent). And during intracellular recordings from HVC in awake, singing birds – every recording is so precious and hard-earned and eyeballed in real time on a computer/oscilloscope that there is no way that pre-bout discharge would have been missed. In these studies, half of the HVC_RA_ neurons were either silent or active during introductory notes (though this was not emphasized in the papers).A real possibility that could be included here is that Ca^2+^ transients do not equal spiking and vice versa. Dopamine 'ramps' are observed in Ca^2+^ imaging of neurons and in fiber photometry but have never been observed in DA spiking (there was a long discussion of this issue at CoSyne 2018 meeting, it is covered in recent review by Josh Berke (Berke, 2018), and is relevant to ramping signals observed in the recently posted biorxiv paper from the Witten group (https://www.biorxiv.org/content/10.1101/456194v1)).Also, inhibition can independently regulate Ca^2+^ influx and spiking – resulting in non-congruence between spikes and GCAMP Ca^2+^ signals (nicely shown in Owen, Berke and Kreitzer, 2018);

Ok. We have modified this section. We now acknowledge at the beginning that we are speculating. We also specifically cite the Hahnloser, 2002 study when we mention methods for making the initial recordings of HVC-RA neurons using short buffering windows. According to Richard Hahnloser, an author on this manuscript and the first author on the 2002 study, audio monitors were not used in the recordings reported in that study and it is not how he has done these types of recordings in the past. Different studies may have been carried out in different ways over the years, so we have only cited the Hahnloser, 2002 paper in this section. Lastly, we have also added the possibility that there could be incongruencies between spiking activity and calcium signals (citing the Kreitzer paper) and acknowledged that the relationship between spiking and calcium signals has not been tested during peri-song epochs in freely behaving birds.

Discussion section: Get rid of 'exclusively'.

OK.

Discussion section: Get rid of 'only'.

OK.

Discussion section: Again, the statement that pre-bout Activity in HVC was not previously observed is not accurate; see Picardo et al., Figure 1.

I could not find the statement the reviewer is referring to. The sentence in the first paragraph of the Discussion section reads:

“Moreover, we find that preparatory activity in HVC_RA_ neurons precedes the pre-bout activity described in other classes of HVC neurons and other portions of the song system previously described in zebra finches (Danish et al., 2017; Goldberg et al., 2010; Goldberg and Fee, 2012; Kao et al., 2008; Rajan, 2018; Roberts et al., 2017; Vyssotski et al., 2016; Williams and Vicario, 1993) and described here in Bengalese finches (Figure 4).”

[Editors' note: further revisions were requested prior to acceptance, as described below.]

There is a remaining issue with interpretation and semantics of what is pre- and post-vocalization activity measured in HVC and the song system relative to prior studies, which is what has been causing confusion to the reviewers. From a semantic understanding, pre-song activity in HVC-RA or any neurons of the song system could be considered 1 ms, 10ms, 100ms, 1s, 10s, or 10 min before vocalizing. In all cases, activity increases occur before vocalizing. All are premotor changes in neural firing. The difference that the authors find is that some neurons continue to fire during the motif or syllable production that has been found in previous studies and others do not, the new finding of this study. These former are what the authors appear to characterize as 'during song neurons', even if they fire 1-500 ms before or after the sound itself. The later fire 500ms to 3,000ms before sound is produced, and some of these shut off during the moments of syllable production. The authors assume that the idea of the 1-500ms range of premotor activity is "baked in", and therefore they call them 'during song production/singing neurons'. But what is baked-in the neuroscience field is that there is premotor activity in motor pathways in the manuscript before sound production or other behavior production. Can the authors see the confusion here?

1) For all the calcium imaging data (Figure 1, Figure 2, Figure 3 and associated supplemental figures), we include all activity in the 5 seconds prior to the bird vocalizing as ‘pre-song’ and all the activity in the 5 seconds following song offset as ‘post-song’. We do not exclude the 500ms period just before and just after vocalization as separate data – it is all included in the pre or post-song.

2) For the multi-unit electrophysiological recordings from HVC (Figure 4) and RA (Figure 5) we exclude the 500ms just before and just after song from the spike rate (Figure 4G) and CV ISI calculations (Figure 5D) of pre and post song data. The exclusion of the 500ms windows is illustrated in panels 4A and 5A. We did this because activity in HVC and RA are known to increase in the last ~300ms prior to song onset and we did not want this activity to bias our measurements at longer times either before or after singing. In the context of this paper we show that HVC_RA_ neurons are active on the order of seconds before singing. We were interested in seeing if we could detect early or late activity using multi-unit recordings in HVC and RA from another species. If we included the 500ms time-period just around the song in these measurements it would make our data look stronger, but it would not correctly address our question. Therefore, we have taken a conservative approach to analyzing this data. The synaptic delays associated with descending motor commands are thought to be on the order of 25-50ms in the songbird system – well within the 500ms time-window excluded from this analysis.

3) Lastly, in our calcium imaging data set from HVC-RA neurons we found less than 4% of activity falling within last 100ms prior to song onset, suggesting that >96.2% of activity described here falls outside of the time-window thought to be involved in providing descending motor commands controlling vocalization. This supports our arguments that HVC-RA neurons play a role in motor planning or preparation rather than merely reflecting synaptic delays associated with singing. We touch on these ideas in the Results section. In subsection “Peri-Song Activity in Populations of HVC_RA_ Neurons” we say: “The sequence of syllables in zebra finch song is stereotyped and unfolds in less than a second. Like other rapid and precise motor movements, song may benefit from motor planning and preparatory activity unfolding on much longer timescales than the synaptic delays associated with descending motor commands, which in zebra finches are estimated to be ~25-50ms (Amador et al., 2013; Fee et al., 2004)” In subsection “Peri-Song Activity in Populations of HVC_RA_ Neurons” we say: “The vast majority of pre-song activity we describe occurs hundreds of milliseconds to seconds prior to vocalizations, suggesting a role in planning or preparation to vocalize (96.2% of calcium events occurred more than 100 ms prior to vocal onset and 65.6% occurred more than 1 second prior to vocal onset).”

To be less confusing to readers, the authors should include a table that defines the different neuron categories according to their temporal firing patterns, and then use those definitions consistently throughout the paper. Example terms that need such definitions in the table are: peri-song neurons (presumably within 5-0.5s firing before or after the first or last song syllable, respectively); pre-song neurons (presumably firing ~0.5s to 1ms before each syllable or the first song syllable); post-song; pan-song, etc.

We now provide Supplementary file 2 with definitions for the three categories of HVC-RA neurons described in this paper.

The real novelty of this study is finding neurons of the peri-song neurons as defined above, and not even just HVC-RA projecting neurons, but the fact that any such neurons exist at all in the song system. The paper needs to highlight this novelty more clearly. To do so, this difference in timing of neuron population firing and pattern should be stated in the Abstract, end of Introduction and in sentences of the results when this topic is discussed.

We believe that we have highlighted these findings in the submission in the Abstract, Introduction and Discussion section.

In the Abstract we say:

“Using cell-type specific calcium imaging we identify populations of HVC premotor neurons associated with the beginning and ending of singing-related neural sequences. We characterize neurons that bookend singing-related sequences and neuronal populations that transition from sparse preparatory activity prior to song to precise neural sequences during singing.”

At the end of the Introduction we say:

We show that ~50% of HVC_RA_ neurons are active during periods associated with preparation to sing and recovery from singing and that their activity presages the volitional production of song by 2-3 seconds. One population of HVC_RA_ neurons is only active immediately preceding and following song production, but not during either singing or non-vocal behaviors. A second population of neurons exhibits ramping activity before and after singing and can also participate in precise neural sequences during song performance.

At the beginning of the Discussion section we say:

“Our principal result is that neural sequences among HVC_RA_ neurons emerge as part of orchestrated population activity across a larger network of HVC_RA_ neurons. This activity can be correlated with motor planning and preparation prior to song initiation. Peri-song and pan-song HVC_RA_ neurons forecast the start of singing. Peri-song HVC_RA_ neurons are inactive during song, whereas pan-song neurons become heterogeneously active prior to time-locked sequential activity during song performances (Figures 2 and 3). Moreover, we find that preparatory activity in HVC_RA_ neurons precedes the pre-bout activity described in other classes of HVC neurons and other portions of the song system previously described in zebra finches (Danish et al., 2017; Goldberg et al., 2010; Goldberg and Fee, 2012; Kao et al., 2008; Rajan, 2018; Roberts et al., 2017; Vyssotski et al., 2016; Williams and Vicario, 1993) and described here in Bengalese finches (Figure 4). This suggests that HVC_RA_ neurons may seed network wide changes among other classes of HVC neurons and the song system more generally.”

We would also like to point out that Rajan, 2018 found that there is a subclass of HVC-X neurons that is active immediately before and after song, but not active during singing. Therefore, we do not feel that identification of any neuron with this firing pattern, as the Reviewing Editor suggests, is the real novelty of this study. We discussed this in detail in the response to our original submission. What we think is truly novel is that some HVC-RA neurons behave in a much more complicated fashion than previously known. We have identified 3 sub-populations of HVC-RA neurons that have activity patterns going well beyond sequencing of motor acts, the activity of two of these populations can start seconds before singing, and that precise sequences for song are embedded within this complex orchestrated pattern of activity. Moreover, the activity of HVC-RA neurons precedes the activity of all the pre-bout activity in the song system that has been reported to date (as far as we know).

More specifically, what appears most novel about the current study is that the authors find subpopulations of HVC-RA projection neurons that: (a) fire 1-3 seconds before singing starts [or after it ends]; (b) some of which turn off a few ms before or right at the time singing starts; and (c) are not correlated with as of yet identified pattern of singing. The other types of neural firing patterns, including pre- and post-song production, have been found in prior studies, and are confirmed in this study. This needs to be stated much more clearly in the manuscript.

We feel like these points have already been addressed in the paper.

In the Introduction we say:

“Neuronal activity related to motor planning and preparation has been associated with accurate production of volitional motor movements(Churchland et al., 2010a; Svoboda and Li, 2017) but is still poorly described in the context of initiating precise neural sequences for motor behaviors, like those exhibited in HVC_RA_ neurons. Although it is not known if HVC_RA_ neurons exhibt activity related to motor planning and preparation, previous studies have identified anticipatory or preparatory activity in other classes of HVC neurons and in other regions of the songbird brain(Goldberg et al., 2010; Goldberg and Fee, 2012; Kao et al., 2008; Keller and Hahnloser, 2009; Rajan, 2018; Roberts et al., 2017). HVC contains interneurons and at least three classes of projection neurons, including neurons projecting to the striatopallidal region Area X (HVC_X_), neurons projecting to a portion of the auditory cortex termed Avalanche (HVC_Av_), and the aforementioned HVC_RA_ neurons that encode precise premotor sequences necessary for song production(Akutagawa and Konishi, 2010; Mooney and Prather, 2005; Roberts et al., 2017). Multi-unit recordings from HVC, which are typically dominated by the activity of interneurons, show increases in activity tens to hundreds of milliseconds prior to singing(Crandall et al., 2007; Day et al., 2009; Rajan, 2018). Calcium imaging from HVC_Av_ neurons and electrophysiological recordings from HVC_X_ neurons indicate that they also become active immediately prior to song onset (Rajan, 2018; Roberts et al., 2017). These data are consistent with recordings from the downstream targets of HVC_Av_ and HVC_X_ neurons. Portions of the auditory cortex (Keller and Hahnloser, 2009) and the basal ganglia pathway involved in song learning show changes in activity immediately prior to singing (Goldberg et al., 2010; Goldberg and Fee, 2012; Kao et al., 2008). Given this background, and that ~50% of HVC_RA_ neurons may not exhibit any activity during singing (Hamaguchi et al., 2016; Long et al., 2010), we sought to examine if the precise neural sequences associated with song arise as part of larger changes in activity among populations of HVC_RA_ neurons.”

Introduction: Making too strong of a statement that HVC-RA neurons are exclusively active during vocalization in awake birds. Even in awake birds, physiology and immediate gene studies have shown some activity to hearing songs in other species (even if birds own song) or eating (even if difficult to interpret gene expression studies). Instead of saying exclusively, how about predominantly.

We have now changed this sentence, so it reads:

“HVC_RA_ neurons are thought to only be active during vocal production in waking adult birds, yet ~50% of recorded HVC_RA_ neurons do not exhibit any activity during singing(Hahnloser et al., 2002; Hamaguchi et al., 2016; Kozhevnikov and Fee, 2007; Long et al., 2010; Lynch et al., 2016), leaving the function of much of the HVC_RA_ circuitry unresolved.”

As we discussed thoroughly in our response to the first round of reviews, there is not any evidence that we have been able to find showing that HVC-RA neurons are active outside of singing in awake adult songbirds. The IEG study referenced by the editor does not show that HVC-RA neurons are active in response to eating, and there are not any electrophysiological recordings of identified HVC-RA neurons from any species showing that they are active outside of song in awake adult birds. We want to be correct and we have extensively searched the literature. If the Reviewing Editor can provide us with literature showing that HVC-RA neurons are active outside of singing in adult waking songbirds we would be happy to change our text, but as it stands, we do not feel that our statement is too strong. Moreover, changing the text to say “predominantly”, as suggested, implies that HVC-RA neurons in adult waking birds are known to be active outside of song.

Results section: The authors claim that the pre-bout burst in HVC is a "newly discovered activity profile." But that's not accurate when considering the 500 ms window of pre-bout activity. Here is where the authors would benefit by using more well-defined terminology that they show in a table as well as define the timing of what they mean by pre-bout activity.

This was corrected in an earlier version of our manuscript and is not included in the current submission.

Discussion section: It appears the Bengalese finch electrophysiology peri-song activity is within a shorter time window of several 100ms rather than 1-3 seconds as found with Ca^2+^ imaging in zebra finches. Is this a species difference, a physiology versus GCAMP sensing difference, or some other difference?

The differences between the zebra finch and Bengalese finch experiments may arise from using multi-unit electrophysiological recordings, which are dominated by the activity of interneurons, versus the cell type specific calcium imaging methods that allowed us to image exclusively from populations of HVC-RA neurons. We find that pre-song activity increased above baseline -1.47 ± 1.05 s prior to song onset in Bengalese finches, a timescale that closely matched the timing of peak calcium-event rates in peri-song neurons in zebra finches. Previous multiunit recordings from zebra finch HVC show elevated activity prior to song onset on the scale of tens to hundreds of milliseconds prior to song onset, while our recordings from Bengalese reveal increased activity ~1.5 sec prior to song onset. Therefore, species differences in preparatory activity at the level of HVC interneurons may also exist. Controlled multi-unit recordings from zebra finches and Bengalese finches would be needed to fully resolve this question.

The song system diagram of Figure 1A is appears to be simplified too much. It does not show MAN, the loop with AreaX and DLM, and the feedback to RA (as well as to HVC). Such additions to the figure would not cause more confusion, and in fact would add clarity and has been what is commonly shown in birdsong diagrams for several decades now. If it would make it easier for the authors, they could use and modify one of the figures showing these connections in the adobe illustrator versions on the Jarvis Lab website: http://jarvislab.net/summary-figure-originals/ These are more polished than the hand drawn one of this paper.

we have added a supplemental figure that includes the entire song system. We prefer to keep Figure 1A as it is, but the supplemental figure will provide interested readers with a complete diagram of the song system.

As the authors wrote in one of their responses, they should mention that the variability of peri-song neuron activity on different bouts is similar to variable pre-movement activity found primate M1 neurons. Such a comparison broadens the impact of the paper.

We added this to the Discussion section in our first resubmission. The second to last paragraph of the Discussion section reads:

“Neural activity associated with motor planning and preparation has been observed in motor and premotor cortices for a variety of different motor tasks in rodents and primates (Chen et al., 2017; Churchland et al., 2010a; Churchland et al., 2006a; Churchland et al., 2006b; Economo et al., 2018; Inagaki et al., 2019; Kaufman et al., 2014; Li et al., 2015; Li et al., 2016; Murakami and Mainen, 2015; Murakami et al., 2014; Tanji and Evarts, 1976), including vocalizations (Gavrilov et al., 2017). This activity is thought to reflect the decision to perform movements and is characterized by a high degree of variability from trial to trial. Changes in circuit dynamics function to shift the initial state of a network to levels that enable efficient and accurate motor performances and this activity can start to unfold seconds prior to movement initiation(Churchland et al., 2010a; Inagaki et al., 2019; Murakami and Mainen, 2015; Svoboda and Li, 2017). HVC is proposed to be analogous to the mammalian motor cortex (layer III neurons of the primary motor cortex) (Pfenning et al., 2014), or premotor cortex(Bolhuis et al., 2010). Our observation of preparatory activity seconds before the onset of courtship song in two different songbird species suggests that pre-movement activity is a common mechanism for ensuring the accurate production of volitional behaviors. In line with recordings of pre-motor activity in mammals, individual HVC_RA_ neurons exhibit a high degree of trial to trial variability. Although zebra finch courtship song is famously stereotyped, there is a measurable degree of variability in the structure and duration of song motifs, bouts and phrases from trial to trial. Recording of preparatory activity across larger populations of HVC_RA_ neurons may be useful for decoding this trial to trial variability in song structure, ultimately providing a predictive readout of impending behaviors.”